# Robust Strategic Classification under Decision-Dependent Cost Uncertainty

**Sura Alhanouti** [1][2]  **Güzin Bayraksan** [1]  **Parinaz Naghizadeh** [3]

## Abstract

Humans facing algorithmic decision systems have been found to "game" them by altering their input data (at a cost to them) in order to favorably change the algorithmic outcomes they receive (at a cost to the algorithm). The growing literature on strategic classification seeks to develop robust machine learning algorithms that account for, and reduce, unwanted strategic behavior. A limitation of these existing works is that they assume the cost of strategic behavior to be fixed and independent of the classifier's decision. In practice, however, manipulation costs evolve and depend on past algorithmic decisions: today's decisions influence tomorrow's costs. This paper proposes and analyzes a two-stage robust optimization framework with a decision-dependent uncertainty set to capture such dependencies. We highlight that awareness of policy-dependent costs not only reduces uncertainty, but also better curtails gaming of the algorithmic system over time.

## 1. Introduction

Machine learning algorithms are increasingly deployed in decision making systems, including in financial lending, job hiring, recidivism prediction, and school admission decisions. A key issue arising in these systems is *strategic behavior* by humans, who modify their inputs to the algorithm (e.g., by misrepresenting their features), in a way that allows them to gain a favorable outcome but does not necessarily align with the system's goal of making accurate predictions (Pramanik, 2025; Maag, 2024; Hadero, 2024; Stahl, 2022; Solman, 2017; Möhlmann & Zalmanson, 2017).

---

[1]Department of Integrated Systems Engineering, The Ohio State University, Ohio, USA [2]Department of Industrial Engineering, Jordan University of Science and Technology, Irbid, Jordan [3]Department of Electrical and Computer Engineering, University of California, San Diego, USA. Correspondence to: Sura Alhanouti <alhanouti.1@osu.edu>, Güzin Bayraksan <bayraksan.1@osu.edu>, Parinaz Naghizadeh <parinaz@ucsd.edu>.

*Proceedings of the 43rd International Conference on Machine Learning*, Seoul, South Korea. PMLR 306, 2026. Copyright 2026 by the author(s).

This has motivated a growing line of works on strategic machine learning, which study the design of algorithms that are robust to such "gaming" of the system.

The majority of this existing literature adopts a limiting assumption: both the decision maker and the agents have full information about the algorithmic system and the strategic recourse options, including which features can be manipulated and at what cost. Recent work has started introducing uncertainty into this problem by considering agents with incomplete or biased perceptions of the classifier (Cohen et al., 2025; Ebrahimi et al., 2025; Haghtalab et al., 2023) or classifiers uncertain about agents' costs (Rosenfeld & Rosenfeld, 2024; Shao et al., 2023; Ahmadi et al., 2021).

However, a type of uncertainty that arises in practice, but remains unaddressed by these works, is that classifier decisions that aim to curb agents' gaming can *endogenously* shape cost uncertainty *over time*. Specifically, the criteria set by the present classifier influence the possible set of costs that the agents face when strategically responding to the algorithmic system in the future. For instance, test-optional policies started by colleges during the COVID-19 pandemic reduced demand and prices for test preparation in subsequent years, while simultaneously shifting spending toward alternative costly signals such as essay coaching and extracurricular activities (Petrosino, 2025; Deming, 2024; Spitalniak, 2023; Edwards, 2022). Thus, a school's decisions this year shape future applicants' strategic response costs; although, crucially, the exact impact is not fully known at the time of decision-making. Similar effects appear in other contexts such as lending and hiring (Gillis & Simpson, 2025; Björkegren et al., 2020; Bambauer & Zarsky, 2018), wherein the criteria adopted for approval have been found to shape the options and costs of strategic gaming available to future applicants. These motivate the need to not only address strategic manipulation, but to do so with an eye towards its uncertain temporal effects.

This paper is, to our knowledge, the first to formalize such *time-evolving and decision-dependent uncertainty* in strategic classification. We formulate the classifier selection as a two-stage robust optimization (TSRO) problem. In the first stage, the classifier anticipates that its choices will influence future response costs by the agents; we model this through a decision-dependent uncertainty set for the second-stage

problem. Throughout, we provide support for different elements of our proposed model using university admissions as a motivating example, including by highlighting universities' awareness of how their admissions' criteria impose long-term costs and shape strategic behavior.

Following the problem formulation, we note that solving the resulting TSRO remains challenging. Specifically, our second stage problem is nonlinear, and the uncertainty appears not only in the constraint set, but also in the objective through a dual-norm term, causing the dual feasible set to vary with each realization and preventing the reuse of dual extreme points or valid cut generation to solve the problem. We address this by proposing problem-specific approximations and reformulations. Specifically, we propose an approximation of the cost-induced norms that appear in the firm's per-stage objective (which we show has an added benefit in the context of strategic classification, by enhancing robustness against harmful manipulations), linearize the objective function, and use McCormick envelopes to remove decision-dependence in the second-stage problem's objective function. Following these, the relaxed problem becomes amenable to some existing solution methods for linear TSROs, including the commonly used Benders decomposition and Column-and-Constraint Generation (C&CG) algorithms (Qiu et al., 2024; Zeng & Wang, 2022; Zhang et al., 2022; Chen & Wei, 2022), which we employ. We validate and bound the suboptimality gap of our approximations numerically by comparing the solution obtained from the relaxed problem to the original TSRO's optimal solution obtained via brute-force search; see Section 6.

We then proceed to using numerical experiments (on both synthetic and semi-synthetic data) to assess the impacts of our method's awareness of time-evolving and decision-dependence uncertainties. Our semi-synthetic experiments are built on real-world datasets on college admission statistics and costs (National Association for College Admission Counseling, 2025; Crimson Education, 2025; Solomon, 2025; Park et al., 2025; Daniel, 2024; College Board, 2023). Our main findings here are two-fold. First, we illustrate that compared to a dependency-unaware baseline, our dependency-aware classifier strategically sacrifices some first-stage performance to gain second-stage robustness; this is a consequence of moving from myopic to temporally-aware decisions. Second, and less expected, we show that our method reduces overall loss not only due to its awareness of temporal effects and robustness to uncertainty, but by changing agents' costs in a way that limits strategic manipulation. In other words, our experiments show that accounting for time-evolving, decision-dependent costs allows a decision maker to not only minimize its uncertainty, but also *shape strategic responses*, highlighting a different (and previously unexplored) lever for mitigating gaming of algorithmic systems over time.

## 2. Related Work

Most of the works studying humans' strategic responses to AI/ML systems assume full information for both classifiers and agents and explore the implications of agents manipulating inputs without changing their true qualifications (Hardt et al., 2016; Milli et al., 2019; Levanon & Rosenfeld, 2022; Sundaram et al., 2023), genuinely improving their qualifications (Ahmadi et al., 2023; Tsirtsis et al., 2024), or choosing between manipulation and improvement (Wang et al., 2023; Horowitz & Rosenfeld, 2023; Haghtalab et al., 2021; Harris et al., 2021; Bechavod et al., 2021; Alhanouti & Naghizadeh, 2024; Efthymiou et al., 2025; Xie et al., 2024).

More recent works examine *agents having incomplete information about the algorithm*. Braverman & Garg (2020), Geary & Gouk (2025), and Sundaram et al. (2023) propose randomized classifiers to obscure decision boundaries. Cohen et al. (2025) and Harris et al. (2022) explore strategic communication. Ghalme et al. (2021) and Bechavod et al. (2022) study opacity and the fairness implications of limited information about the algorithm.

A separate line of work, similar to ours, handles *firms' uncertainty about agent responses*. These include learning under unknown manipulations (Shao et al., 2023; Lechner et al., 2023; Cohen et al., 2024), unknown costs or preferences (Ahmadi et al., 2021; Dong et al., 2018), or distributional shifts (He et al., 2025). Some models address robustness to uncertain manipulation costs using worst-case optimization or distributionally robust methods (Rosenfeld & Rosenfeld, 2024; Tang et al., 2021).

Our work differs from, and complements, these existing works by considering a new form of incomplete information: a firm's uncertainty over *endogenously evolving manipulation costs*, where future gaming costs depend on the classifier's current decisions. Our proposed form of information asymmetry is supported by existing studies showing how decision rules endogenously shape manipulation incentives; e.g., lax oversight or platform design choices can reduce manipulation costs and increase gaming (Jacob & Levitt, 2003; Alcobendas & Zeithammer, 2022). The idea of classifiers *endogenously* impacting costs in strategic classification has been recently explored by Sommer et al. (2025), who model the classifier's impact on the market demand for feature-altering services. This model captures *immediate* endogenous effects, assuming the classifier anticipates the *current* price equilibrium. Our work differs in two key respects: first, it models *future* costs as being endogenously impacted by current decisions; and second, it accounts for the *uncertainty in this effect*, which is critical in the identified application domains. To the best of our knowledge, this is the first work that formally models and analyzes the consequences of such *uncertain* and *endogenously evolving* dependencies in algorithmic classification.

## 3. Problem Setting and Preliminaries

We study a problem of strategic binary classification, where a firm makes accept/reject decisions on agents with observable features $x \in \mathcal{X} \subseteq \mathbb{R}^d$ and unobservable true labels $y \in \mathcal{Y} = \{\pm 1\}$. We let $(X, Y) \sim P_{XY}$, where $P_{XY}$ denotes the joint distribution over $\mathcal{X} \times \mathcal{Y}$. A lowercase $(x, y)$ denotes a realization of $(X, Y)$. The classification problem unfolds over two stages, with a long-lived firm and short-lived agents (i.e., the same firm makes decisions over both stages, but for a different set of agents each time).

**The first stage.**  At the first stage, the firm selects and announces a linear binary classifier $\text{sign}(\beta^T x)$, where the classifier weights $\beta$ may be restricted to some set $\mathcal{B}$. First-stage agents can strategically respond to classifier $\beta$ by altering their original features $x$ in a way that leads to them being admitted by the classifier, without altering their true labels. Note that this type of strategic response includes both harmless manipulations (e.g., hiring a tutor to learn SAT test tricks) and harmful manipulations (e.g., cheating on the SATs), though in both cases we assume the true qualification state (e.g., ability to graduate) remains unchanged. Formally, a starting feature $x$ will be changed to

$$\hat{x}(\beta) := \arg\max_{\hat{x} \in \mathcal{X}} [\mathbb{1}(\beta^T \hat{x} \geq 0)u - c(x, \hat{x})], \quad (1)$$

where $u \geq 0$ is the utility gained from positive classification, and $c(x, \hat{x})$ denotes the cost of the strategic response. Note that an agent opts for a strategic response only if the change shifts the classification from negative to positive and the cost of achieving this satisfies $c(x, \hat{x}) < u$. Otherwise, the agent takes no action.

**The cost of strategic responses.**  We consider cost functions of the form $c(x, \hat{x}) = \phi(\|\hat{x} - x\|_\Sigma)$, where $\phi : \mathbb{R}_{\geq 0} \to \mathbb{R}_{\geq 0}$ is a non-decreasing function and

$$\|\hat{x} - x\|_\Sigma := \|\Sigma^{\frac{1}{2}}(\hat{x} - x)\|, \quad (2)$$

where $\Sigma \in \mathbb{R}^{d \times d}$ is a positive definite (PD) *cost matrix* ($\Sigma \succ 0$) which uniquely parametrizes the cost function, and $\|\cdot\|$ is the standard $p$-norm with $p \geq 1$. This cost function was proposed by Rosenfeld & Rosenfeld (2024). We note that setting $\Sigma = I_{d \times d}$ recovers the previously studied $\ell_p$-norm costs as a special case (Cohen et al., 2025; Trachtenberg & Rosenfeld, 2025; Efthymiou et al., 2025), including the widely used $\ell_2$-norm (Haghtalab et al., 2021; Horowitz & Rosenfeld, 2023; Ebrahimi et al., 2025).

For additional intuition about this function, note that if $\Sigma$ is diagonal, i.e., $\Sigma = \text{diag}(\sigma_1, \sigma_2, \ldots, \sigma_d)$, then $\sigma_i$ controls the relative cost of modifying feature $x_i$, with larger $\sigma_i$ indicating higher cost. If $\Sigma$ has off-diagonal entries, it encodes correlations in feature change costs: changing feature $x_i$ may influence the cost of changing another feature $x_j$.

We assume that in the first stage, the exact cost of strategic responses $c(x, \hat{x})$ by the first-stage agent population is known to the firm, and is parameterized by a cost matrix $\Sigma_0$. The firm can use this knowledge to anticipate the impact of agents' strategic responses when it is choosing its classifier $\beta$. This stage, in isolation, aligns with the classic strategic classification problem studied in prior works (e.g., Cohen et al. (2025); Haghtalab et al. (2021); Ahmadi et al. (2021)), but with a more general cost function.

**The second stage.**  Our model differs from existing strategic classification frameworks in its second stage and its coupling with the first stage. Crucially, the firm anticipates that first-stage decisions affect the cost of second-stage strategic responses, although the impact is uncertain.

Formally, we let the second-stage cost for changing features from $x$ to $\hat{x}$ be given by

$$c(x, \hat{x}) = \phi\left(\|\hat{x} - x\|_{\Sigma(\omega)}\right). \quad (3)$$

That is, the second stage has an uncertain cost matrix $\Sigma(\omega) \in \mathbb{R}^{d \times d}$. This matrix is assumed to be a function of a random vector $\omega \in \mathbb{R}_+^d$ whose components will capture how the second-stage cost depends on the first-stage decision $\beta$.

Specifically, we define a *decision-dependent uncertainty set* $\Omega(\beta)$, from which the random vector $\omega$ will be drawn:

$$\omega \in \Omega(\beta). \quad (4)$$

During the first stage, the firm's uncertainty about the second stage is parametrized by this set, containing all possible realizations of the cost-driving vector $\omega$ that could arise given the first-stage decision $\beta$. (We formally detail the firm's model of this uncertainty set in Section 4.2.)

At the beginning of the second stage, a realization $\omega$ is drawn from this set, which in turn realizes the cost matrix for the second stage according to

$$\Sigma(\omega) := Q(g(\omega)) \cdot \Sigma_0, \quad (5)$$

where $\Sigma_0 \in \mathbb{R}^{d \times d}$ is the first-stage cost matrix, and $Q(g(\cdot)) \in \mathbb{R}^{d \times d}$ is a matrix-valued transformation that maps the element-wise scaling factors $g(\omega_j)$ into a full cost-scaling matrix. Each component $\omega_i$ controls the scaling of the cost for feature $x_i$ and may also influence the cost of other features $x_j$ with $j \neq i$. To keep the structure tractable, we require $Q(g(\omega))$ to be *component-wise separable* in $\omega$, meaning each entry is a function of a single $\omega_i$ and contains no component product transformations. This ensures $\omega$ influences the cost geometry only through independent scaling determined by $g(\omega)$. Each scaling function $g(\cdot)$ is itself assumed to be bounded and strictly positive (i.e., $0 < g(\cdot) < \infty$), ensuring that $\Sigma(\omega) \succ 0$, and with $g(\omega)^{-\frac{1}{2}}$ linear in $\omega$. (See Appendix D.1 for motivation.)

While our proposed algorithm and analytical results will be applicable to any PD matrix $\Sigma_0$ and functions $Q(\cdot)$ and $g(\cdot)$ satisfying the conditions stated above, identifying such functions is non-trivial, and particularly in practice, real-world cost evolution functions may not meet such admissible forms. As a result, the applicability of our framework will require approximations to match these admissible criteria. Below, we identify one class of PD matrices $\Sigma_0$ and functions $Q(\cdot)$ and $g(\cdot)$ satisfying the conditions stated above, which can be used as a basis for approximating cost evolution functions learned from real-world historical data to fit our proposed framework.

*Example (special case):* Assume $Q(g(\omega))$ and $\Sigma_0$ are both diagonal. This means that $\sigma_i$, the first-stage unit cost of altering feature $x_i$, will change to $g(\omega_i)\sigma_i$ in the second-stage. Specifically, choosing $g(\omega) = \omega^{-2}$ will lead to the second-stage unit costs $\omega_i^{-2}\sigma_i$ scaling inversely with the (squared) realization of the uncertainty parameter. We will use this special case in our numerical experiments in Section 6. We also provide additional experiments on off-diagonal instances in Appendix E.2.

In summary, the second-stage cost depends on the first-stage decision $\beta$ through the following chain of mappings:

$$\beta \mapsto \Omega(\beta) \ni \omega \mapsto \Sigma(\omega) \mapsto \|\cdot\|_{\Sigma(\omega)} \mapsto c(x, \hat{x}).$$

Note that once the second stage begins, a linear classifier $\mathrm{sign}(\beta'^T x)$ is chosen from some set $\mathcal{B}'$, *after* the realization of the decision-dependent costs (i.e., second-stage agents' costs becomes known to the firm at the second stage). It is only the choice of the first-stage classifier $\beta$ that has to be made to restrict gaming in the first stage *while at the same time* anticipating its uncertain impacts on the agents' costs in the second stage. Before formalizing the firm's problem, we provide a motivating example for our proposed model.

**Motivating example: university admissions.** Empirical evidence suggests that shifting to test-optional policies alters the market for preparatory services (such as tutoring for standardized tests) while increasing emphasis on other components like extracurricular activities, which in turn affects the costs of modifying each application component for students (Spitalniak, 2023; Petrosino, 2025). In this context, the firm (here, a university) can observe the current costs students incur to meet that year's admission criteria; this aligns with our assumption that the costs are known to the firm at the time of decision-making. The school also understands that its decisions during the admission season this year (i.e., the weight placed on different application components) will shape the strategic cost landscape for applicants in the following year; this is because this year's applicants use services such as Naviance or websites such as CollegeData Admissions Tracker to learn about the overall application profiles of last year's admitted students. Although the exact

nature of next year's costs is uncertain, the school can incorporate this decision-dependent uncertainty into its current admission policies to better anticipate long-term effects.

This awareness is not merely theoretical. Elite institutions have explicitly acknowledged that their admissions' criteria impose long-term costs and shape strategic behavior. For instance, students often report arriving academically exhausted due to the widespread belief that excessive AP coursework is necessary for admission (McGann, 2006). These reflections, echoed in the "Turning the Tide" report (Harvard Graduate School of Education, 2016), illustrate institutional awareness that current admission signals can fuel competitive overextension and inequitable cost burdens on future applicants. This institutional perspective is reinforced by empirical evidence: Grau (2018) finds that admission policies directly influence high school students' academic effort, with schools emphasizing particular criteria, such as GPA or extracurricular achievement, inducing students to reallocate effort and preparation to align with those priorities.

## 4. The Firm's Optimization Problem

To capture the two-stage decision-making process outlined in the previous section, we formulate the firm's decision process as a two-stage robust optimization problem with a decision-dependent uncertainty set. We formalize this problem throughout this section.

### 4.1. The learning objective

First, consider the learning objective of a single-stage binary classification problem, with a known and fixed strategic cost function. Recall first that agents can opt to strategically respond to a classifier $\beta$ by altering their features from $x$ to $\hat{x}(\beta)$, as shown in (1). Let $\delta_{\Sigma}(x; \beta) := \hat{x}(\beta) - x$ denote the agent's feature change vector in response to classifier $\beta$, when the cost function $c(x, \hat{x})$ is parametrized by a cost matrix $\Sigma$ as shown in (2). Note also that $\delta_{\Sigma}(x; \beta)$ can be an all-zeros vector if the agent takes no action.

The firm's learning objective is to choose a classifier $\beta$ to minimize the expected 0-1 loss while anticipating such strategic responses. Formally, the 0-1 loss incurred on a given strategic agent $(x, y)$ facing cost matrix $\Sigma$ is

$$\ell_{\Sigma}(\beta^{\top} x, y) := \mathbb{1}\{\mathrm{sign}(\beta^{\top}(x + \delta_{\Sigma}(x; \beta))) \neq y\}. \quad (6)$$

Efficient optimization of the (expected) loss in (6) is hindered by two issues: non-convexity from the $\mathrm{sign}$ function and discontinuities in $\delta_{\Sigma}(x; \beta)$. The first admits standard surrogate losses (e.g., hinge), and the second has been tackled by prior work (Levanon & Rosenfeld, 2022; Rosenfeld & Rosenfeld, 2024) using a *cost-aware strategic hinge loss*:

$$\ell_{\Sigma, \text{s-hinge}}(\beta^{\top} x, y) := \left(1 - y(\beta^{\top} x + u_* \|\beta\|_{*, \Sigma})\right)_+, \quad (7)$$

where $(a)_+ := \max\{0, a\}$, the quantity $u_*$ denotes the largest value satisfying $\phi(u_*) \leq u$, and $\|\beta\|_{*,\Sigma} := \sup_{\|v\|_\Sigma = 1} \beta^\top v = \|\Sigma^{-\frac{1}{2}}\beta\|_*$ is the $\Sigma$-*transformed dual norm* of $\beta$, with $\|\cdot\|_*$ denoting the dual norm.

We will adopt this cost-aware strategic hinge loss as the learning function of the firm within both first and second stages, where we set the matrix $\Sigma$ in (7) to $\Sigma_0$ in the first stage, and to $\Sigma(\omega)$ in the second stage. More specifically, we define the empirical risk of a classifier $\beta$ when facing a cost function with cost matrix $\Sigma$ by

$$R_\Sigma(\beta) := \frac{1}{N} \sum_{i=1}^N \left( 1 - y_i \left( \beta^\top x_i + u_* \|\beta\|_{*,\Sigma} \right) \right)_+ . \quad (8)$$

The classifier $\beta$ will be selected by the firm (in part) to minimize this empirical risk. Here, taking the expectation of (7) has been replaced with the empirical loss, which can be assessed based on training data.

## 4.2. Modeling the uncertainty set

We propose to model the *decision-dependent uncertainty* set $\Omega(\beta)$ as a polyhedral set defined by linear inequalities:

$$\Omega(\beta) = \{\omega \in \mathbb{R}_+^d : \mathbf{F}(\beta)\omega \leq \mathbf{h} + \mathbf{G}\beta\}, \quad (9)$$

where $\omega$ represents the cost-driving vector (uncertain variable), and $\beta$ denotes the classifier's first-stage decision vector. Here, $\mathbf{h}$ serves as a baseline constraint (e.g., capturing average budgets or behavioral limits) while the matrices $\mathbf{F}(\beta)$ and $\mathbf{G}$ capture how the decision $\beta$ influences the structure and bounds of the uncertainty set, respectively, and thus its influence on the feasible magnitude of manipulation incentives. Note that if $\mathbf{F}$ is a fixed ($\beta$-independent) matrix, the inequalities constrain each cost component independently. In contrast, allowing $\mathbf{F}$ to depend on $\beta$ means that the classifier can alter how different cost components interact. In practice, each component of the matrices can be set using domain expertise or estimated from historical data on strategic behavior; see Appendix B for more discussion.

**Motivating example, continued.** In college admissions, shifting emphasis from standardized tests to GPA or extracurriculars can alter the strategic cost landscape. For example, de-emphasizing SAT scores (e.g., through test-optional policies) has decreased demand for test prep (lowering SAT-related costs), while increasing demand for tutoring in coursework or essay coaching (Spitalniak, 2023; Petrosino, 2025), thus raising those prices. The constraint $\mathbf{F}(\beta)\omega \leq \mathbf{h} + \mathbf{G}\beta$ captures this shift: $\mathbf{h}$ can represent a tutorial class's average price; the term $\mathbf{G}\beta$ reflects cost adjustments driven by the relative importance of different admissions criteria; $\mathbf{F}(\beta)$ governs the structural relationships among cost components, altering how increases in one type of preparation (e.g., GPA-related tutoring) constrain or interact with others (e.g., extracurricular coaching).

The polyhedral set form of (9) aligns with standard constructions commonly used in the literature (e.g., Qiu et al., 2024; Zeng & Wang, 2022; Zhang et al., 2022). Its choice facilitates the computational tractability of TSRO problems with decision-dependent uncertainty by allowing an efficient reformulation of the TSRO problem as a linear program. Specifically, with this choice, we can solve the inner maximization over the uncertainty set $\Omega(\beta)$ by leveraging KKT conditions, enabling iterative cut generation over a finite fixed set of scenarios in the outer loop. Further algorithmic details are provided in subsequent sections.

## 4.3. The two-stage robust optimization problem

We are now ready to state the firm's two-stage robust optimization (TSRO) problem with the proposed decision-dependent uncertainty set. The firm solves the following nested min-max-min problem in the first stage:

$$\min_{\beta \in \mathcal{B}} \left( R_{\Sigma_0}(\beta) + \max_{\omega \in \Omega(\beta)} \min_{\beta' \in \mathcal{B}'(\beta, \omega)} R_{\Sigma(\omega)}(\beta') \right), \quad (10)$$

where $R_\Sigma(\beta)$ are the empirical risks, as defined in (8), and $\Omega(\beta)$ is the decision-dependent cost uncertainty set defined in (9). In words, the firm wants to choose the classifier $\beta \in \mathcal{B}$ by **minimizing** the strategic hinge loss under a known fixed cost $\Sigma_0$, while accounting for the worst-case (**maximum**) loss over the possible cost functions $\Sigma(\omega)$ induced in the second stage by the choice of $\beta$, and accounting for the fact that the second-stage classifier $\beta' \in \mathcal{B}'$ is itself selected to **minimize** the second stage's loss after $\omega$ is realized. (The bold terms in this description align, in order, with the min-max-min operations in (10).)

We also note that the choice of classifiers can be restricted, if desired, to specific sets. Formally, $\mathcal{B} = \{\beta \in \mathbb{R}^d : \mathbf{A}\beta \geq \mathbf{b}\}$ represents the feasible set for the first-stage classifier, which, *when needed*, can encode constraints related to fairness, feature importance, or interpretability. For example, in the context of university admissions, if an institution places restrictions on the weight assigned to GPA, or on interaction terms between GPA and essay scores, these requirements can be modeled through the constraint matrix $\mathbf{A}$ and vector $\mathbf{b}$. Similarly, the second-stage feasible set, $\mathcal{B}'(\beta, \omega) = \{\beta' \in \mathbb{R}^d : \mathbf{B}_2\beta' \geq \mathbf{d} - \mathbf{B}_1\beta - \mathbf{E}\omega\}$, defines the space of allowable adjustments to the classifier in the second stage. This set can also be instantiated *when needed* to reflect downstream policy or resource constraints. For example, in school admissions, suppose that shifting emphasis away from standardized tests leads to a surge in costly extracurricular investments by applicants. In response, the school may wish to limit the weight placed on extracurriculars in the subsequent year, either to mitigate inequity or to comply with institutional policy. Such adjustments (e.g., enforcing minima or maxima on feature weights) can be encoded via the matrices $\mathbf{B}_1, \mathbf{B}_2$, and $\mathbf{E}$, capturing how

second-stage policy decisions $\beta'$ will depend on prior decisions and observed cost shifts.

# 5. Solving the Firm's Optimization Problem: Approximations and Reformulations

In this section, we present an algorithm to solve the firm's two-stage robust optimization problem in (10). Although there exist a number of recent methods for solving TSROs with decision-dependent uncertainty (see Section 1), these methods are not directly applicable to our proposed model due to differing structural requirements.

Specifically, our second stage problem in (10) is nonlinear, and the uncertainty appears not only into the constraint set, but also in the objective through the dual-norm term $\|\beta'\|_{*,\Sigma(\omega)}$, causing the dual feasible set to vary with each realization and preventing the reuse of dual extreme points or valid cut generation. To address this, we approximate the dual-norm with a simpler expression that removes the nonlinearity, but introduces bilinear terms involving $\omega$ and $\beta'$. We then reformulate the problem using additional variables to linearize the max operator in the strategic hinge loss, and apply the McCormick envelope method to relax the bilinear terms, yielding a linear second-stage problem. These approximations and reformulation make the resulting problem compatible with standard C&CG-based solution methods for TSROs. In the remainder of this section, we provide details on the above reformulation steps, along with intuitive support for them in the context of strategic classification.

## 5.1. Approximating the dual-norm $\|\cdot\|_{*,\Sigma}$

Consider the empirical risk (8) appearing in the firm's optimization problem. This risk contains terms of the form $\|\beta\|_{*,\Sigma}$, due to which the cost matrix $\Sigma(\omega)$ introduces an input-dependent geometry affecting the feasible region of the dual problem. Reformulating this norm is therefore the first step in the process, ensuring that the second-stage problem satisfies the structural requirements outlined above for bilevel decomposition and tractable optimization. Specifically, we propose the following bound.

**Lemma 5.1.** *There exists a constant $M > 0$ such that*

$$\|\beta\|_{*,\Sigma(\omega)} \leq M\|\Sigma(\omega)^{-\frac{1}{2}}\|_\infty \|\beta\|_\infty.$$

The proof is provided in Appendix C, and invokes the equivalence of norms theorem, and the submultiplicativity of the $\ell_\infty$ norm. We note that $M$ will depend on the $p$-norm adopted in the cost function; Appendix C gives an example.

The upper bound in Lemma 5.1 provides a linearizable surrogate for the dual norm, which we will use to reformulate the second stage problem with a fixed dual feasible set suitable for decomposition. While this bounding argument is strictly necessary only for the second stage (since the cost matrix

$\Sigma(\omega)$ depends on the unknown realization $\omega$), we will also apply the same approximation to the first-stage term. Doing so ensures a consistent treatment of both stages and simplifies the subsequent linearization, making the overall problem more amenable to bilevel decomposition.

**Intuitive support for approximating the dual norm.** Technically, the proposed dual-norm approximation makes the second-stage uncertainty set decision-independent and improves robustness in strategic classification. As shown in Lemma 2.3 in Rosenfeld & Rosenfeld (2024), for a cost function of the form $c(x,\hat{x}) = \phi(\|\hat{x}-x\|_{\Sigma(\omega^k)})$, the quantity $\|\beta'\|_{*,\Sigma(\omega^k)}$ represents the maximum possible score change resulting from an agent's strategic response to a fixed classifier $\beta'$. By approximating this term via the inequality in Lemma 5.1, we obtain a conservative upper bound on the cost-aware strategic hinge loss. For negatively labeled (unqualified) agents, $y_i = -1$, the loss is *upper bounded* as $\ell_{\Sigma(\omega),\text{s-hinge}}(\beta'^\top x_i, y_i) \leq \left(1 + (\beta'^\top x_i + u_* M\|\Sigma(\omega)^{-\frac{1}{2}}\|_\infty \|\beta'\|_\infty)\right)_+$. For positively labeled (qualified) agents, $y_i = 1$, the same approximation gives a *lower bound* on the hinge loss: $\ell_{\Sigma(\omega),\text{s-hinge}}(\beta'^\top x_i, y_i) \geq \left(1 - (\beta'^\top x_i + u_* M\|\Sigma(\omega)^{-\frac{1}{2}}\|_\infty \|\beta'\|_\infty)\right)_+$.

This "asymmetry" has important implications: the approximation leads to a more conservative (i.e., robust) response to the manipulative behavior of unqualified agents, who reduce classifier accuracy by altering features to obtain a false positive. At the same time, it is less conservative toward strategic behavior by already-qualified agents, whose feature manipulation only reinforces the correct decision. Therefore, this approximation supports designing classifiers that prioritize robustness against harmful manipulation, which aligns with real-world settings where accepting unqualified agents carries higher cost than misclassifying qualified ones.

## 5.2. Linear reformulation of the stage problems

Using the proposed bound on the dual norm in Lemma 5.1, we can approximate the firm's optimization (10) as

$$\min_{\beta \in \mathcal{B}} \left[ \frac{1}{N} \sum_{i=1}^N \left(1 - y_i(\beta^\top x_i + u_* M\|\Sigma_0^{-\frac{1}{2}}\|_\infty \|\beta\|_\infty)\right)_+ \right]$$
$$+ \max_{\omega \in \Omega(\beta)} \min_{\beta' \in \mathcal{B}'(\beta,\omega)}$$
$$\left[ \frac{1}{N} \sum_{i=1}^N \left(1 - y_i(\beta'^\top x_i + u_* M\|\Sigma(\omega)^{-\frac{1}{2}}\|_\infty \|\beta'\|_\infty)\right)_+ \right]. \quad (11)$$

Two further issues must be addressed to make (11) amenable to C&CG-based solution methods for TSROs: (i) nonlinear objective function (note the infinity norms and the max operator), and (ii) the second stage objective is decision-dependent (note the appearance of the matrix $\Sigma(\omega)$). We address these issues next.

**Linearizing and removing decision-dependence in the second-stage objective function.** We begin with the second-stage objective function. First, the max operator $((\cdot)_+)$ in the objective function is handled via the introduction of the variables $s_{2,i} \in \mathbb{R}_+$, $i \in [N] := \{1, \ldots, N\}$ to index data points, subject to the constraints $s_{2,i} \geq 0$, and $s_{2,i} \geq 1 - y_i(\beta'^\top x_i + u_* M \|\Sigma(\omega)^{-\frac{1}{2}}\|_\infty \|\beta'\|_\infty)$. Let $s_2 = (s_{2,1}, \ldots, s_{2,N}) \in \mathbb{R}_+^N$ denote the vector of these new variables at the second-stage.[1] Next, we linearize the $\infty$-norm terms by introducing the variables $t_2^q, t_2^\omega \in \mathbb{R}_+$ and imposing the following constraints: $-t_2^q \leq \beta'_j \leq t_2^q$, $\forall j \in [d]$ and $-t_2^\omega \leq \sum_{r=1}^d [\Sigma(\omega)^{-\frac{1}{2}}]_{jr} \leq t_2^\omega$, $\forall j \in [d]$, where we use $[d] := \{1, \ldots, d\}$ to index coordinates of $d$-dimensional vectors. Linearizing the infinity norms introduces a bilinear term in the $s_{2,i}$ variables' constraints; specifically,

$$s_{2,i} \geq 1 - y_i(\beta'^\top x_i + u_* M t_2^\omega \cdot t_2^q).$$

To preserve linearity, we adopt the McCormick envelope (McCormick, 1976), introducing an auxiliary variable $z \in \mathbb{R}_+$ to represent the bilinear term $z = t_2^q \cdot t_2^\omega$, along with the corresponding envelope constraints. This relaxation is tight for bounded variables and widely used in nonlinear programming (Tawarmalani & Sahinidis, 2005; Selvi et al., 2023; Zhen et al., 2021). McCormick envelope requires bounds on $t_2^q$ and $t_2^\omega$. Intuitively, $\|\Sigma(\omega)^{-\frac{1}{2}}\|_\infty$ reflects the most cost-efficient direction of manipulation, while $\|\beta'\|_\infty$ measures classifier sensitivity; thus, their product captures the worst-case strategic impact on the firm loss. These bounds can be derived from domain knowledge. We denote these bounds by $t_{2,\max}^\omega$ and $t_{2,\max}^q$, respectively. Finally, we decompose the second-stage classifier's weight vector as $\beta' = \beta'^+ - \beta'^-$, where $\beta'^+, \beta'^- \in \mathbb{R}_+^d$ denote its positive and negative components. The linearized second-stage problem will be

$$\min_{\beta'^+, \beta'^-, s_2, t_2^q, t_2^\omega, z} \frac{1}{N} \sum_{i=1}^N s_{2,i} \tag{12}$$

subject to

$$s_{2,i} \geq 1 - y_i\big((\beta'^+ - \beta'^-)^\top x_i + u_* Mz\big), \forall i \in [N], \quad ((12).\text{a})$$

$$-t_2^q \leq \beta'^+_j - \beta'^-_j \leq t_2^q, \quad \forall j \in [d], \quad ((12).\text{b})$$

$$-t_2^\omega \leq \sum_{r=1}^d [\Sigma(\omega)^{-\frac{1}{2}}]_{jr} \leq t_2^\omega, \quad \forall j \in [d], \quad ((12).\text{c})$$

$$\mathbf{B}_2(\beta'^+ - \beta'^-) \geq \mathbf{d} - \mathbf{B}_1(\beta^+ - \beta^-) - \mathbf{E}\omega, \quad ((12).\text{d})$$

$$t_2^q \leq t_{2,\max}^q, \quad t_2^\omega \leq t_{2,\max}^\omega, \quad ((12).\text{e})$$

$$z \leq t_{2,\max}^\omega t_2^q, \quad z \leq t_{2,\max}^q t_2^\omega, \quad ((12).\text{f})$$

$$z \geq t_{2,\max}^q t_2^\omega + t_{2,\max}^\omega t_2^q - t_{2,\max}^\omega t_{2,\max}^q, \quad ((12).\text{g})$$

$$\beta'^+_j, \beta'^-_j, s_{2,i}, t_2^q, t_2^\omega, z \geq 0, \ \forall i \in [N], \forall j \in [d]. \quad ((12).\text{h})$$

[1]We note that despite this reformulation adding $N$ (number of sample) constraints, this will not raise computational issues as the algorithm solves this problem in minibatches.

**Linearizing the first-stage objective functions.** While not strictly necessary, we also linearize the first-stage objective function of (11) for consistency and to reduce computation costs. This involves linearizing the max operator with the variables $s_1 = (s_{1,1}, \ldots, s_{1,N}) \in \mathbb{R}_+^N$, linearizing the infinity norm on $\beta$ with the variable $t_1 \in \mathbb{R}_+$, and representing the classifier's weight vector as $\beta = \beta^+ - \beta^-$, where $\beta^+, \beta^- \in \mathbb{R}_+^d$ denote the positive and negative components. The resulting linearized first-stage problem is

$$\min_{\beta^+, \beta^-, s_1, t_1} \frac{1}{N} \sum_{i=1}^N s_{1,i} \tag{13}$$

subject to

$$s_{1,i} \geq 1 - y_i\Big((\beta^+ - \beta^-)^\top x_i + u_* M \|\Sigma_0^{-\frac{1}{2}}\|_\infty t_1\Big), \forall i \in [N], \quad ((13).\text{a})$$

$$-t_1 \leq \beta^+_j - \beta^-_j \leq t_1, \quad \forall j \in [d], \quad ((13).\text{b})$$

$$\mathbf{A}(\beta^+ - \beta^-) \geq \mathbf{b}, \quad ((13).\text{c})$$

$$\beta^+_j, \beta^-_j, t_1, s_{1,i} \geq 0, \quad \forall i \in [N], \ \forall j \in [d]. \quad ((13).\text{d})$$

### 5.3. Final reformulation of the firm's problem

The final, tri-level linear reformulation of our original problem (10) is

$$\min_{v_1 \in \mathcal{S}_{t1}} \quad \frac{1}{N} \sum_{i=1}^N s_{1,i} + \max_{\omega \in \Omega(\beta)} \min_{v_2 \in \mathcal{S}_{t2}(\omega)} \frac{1}{N} \sum_{i=1}^N s_{2,i}. \tag{14}$$

Here, the first-stage and second-stage decision vectors are $v_1 := (\beta^+, \beta^-, s_1, t_1)$ and $v_2 := (\beta'^+, \beta'^-, s_2, t_2^\omega, t_2^q, z)$, respectively, and the feasible sets for the first-stage and second-stage problems are given by $\mathcal{S}_{t1} = \{\beta^+, \beta^-, s_1, t_1 | \ ((13).\text{a}) - ((13).\text{d})\}$ and $\mathcal{S}_{t2}(\omega) = \{\beta'^+, \beta'^-, s_2, z, t_2^q, t_2^\omega | \ ((12).\text{a}) - ((12).\text{h})\}$.

To solve this reformulated problem, we adopt a C&CG-based solution approach, instantiating it using the Benders C&CG algorithm of Zeng & Wang (2022). The application of this algorithm is subject to mild assumptions on (14), which we also assume: (i) For any $\beta \in \mathcal{B}$, $\Omega(\beta) \neq \emptyset$; (ii) $\Omega(\beta)$ is bounded, i.e., for any $\beta \in \mathcal{B}$, $\omega_j < \infty \ \forall j$; (iii) The program $\min\{\frac{1}{N} \sum_{i=1}^N s_{1,i} + \frac{1}{N} \sum_{i=1}^N s_{2,i} : v_1 \in \mathcal{S}_{t1}, \omega \in \Omega(\beta), v_2 \in \mathcal{S}_{t2}(\omega)\}$ has a finite optimal value.

C&CG methods iteratively refine a master problem and generate cuts based on worst-case uncertainty realizations. Our earlier linearizations and norm approximations are crucial, as they allow us to recast the original nonlinear tri-level formulation (10) into the linear tri-level reformulation (14), which can then be reformulated into a linear bilevel problem and ultimately into a form suitable for C&CG-based methods. Details of the master problem, subproblem formulations, and algorithmic steps appear in Appendices D.1–D.3.

*Table 1.* Average $\pm$ standard error across stages and totals.

| Metric | First-stage DD | First-stage DI | Second-stage DD | Second-stage DI | Total DD | Total DI |
|---|---|---|---|---|---|---|
| 0-1 Loss | $26.80 \pm 0.92$ | $25.92 \pm 0.79$ | $7.43 \pm 0.36$ | $23.74 \pm 3.79$ | **34.23** | 49.66 |
| Manipulations | $37.00 \pm 1.10$ | $33.84 \pm 0.79$ | $2.50 \pm 0.40$ | $18.33 \pm 3.79$ | **39.50** | 52.17 |
| Qualified Manip. | $23.52 \pm 0.98$ | $9.12 \pm 0.49$ | $0.18 \pm 0.03$ | $1.84 \pm 0.79$ | **23.70** | 10.96 |
| Unqualified Manip. | $13.44 \pm 0.67$ | $24.64 \pm 0.75$ | $2.30 \pm 0.38$ | $16.46 \pm 3.34$ | **15.74** | 41.10 |

## 6. Numerical Experiments

We now evaluate our two-stage robust strategic classification framework with decision-dependent uncertain manipulation costs by solving the reformulation obtained in (14). We conduct experiments on both synthetic (see Appendix E.6) and real-world university admission (semi-synthetic) datasets.

We consider a university admissions setting with two features: $x_1$ (SAT score) and $x_2$ (number of extracurricular activities). SAT distributions are taken from the SAT Suite of Assessments Annual Report (College Board, 2023) and extracurricular counts from (Park et al., 2025), both for 2018–2019. Each feature is standardized, and samples are drawn from the respective empirical distributions. Admissions weights follow survey data (National Association for College Admission Counseling, 2025): 0.85 on SAT and 0.15 on extracurriculars. Labels are assigned as $y = 1$ if $0.85x_1 + 0.15x_2 > \theta$ and $y = -1$ otherwise, with $\theta$ defined by a 75th-percentile SAT score (1210) and a moderate extracurricular profile (4–5 activities) (Crimson Education, 2025; Park et al., 2025). To model noise, we adopt feature-dependent mislabeling (Zhang et al., 2021; Smart & Carneiro, 2023), with mislabeling more likely near the boundary. Through this process, we generate a dataset of 10,000 samples. See Appendix E.1.1 for more details.

Students receiving negative outcomes may strategically manipulate their features to obtain a positive outcome, and receive utility $u = 1$ in both stages. The manipulation cost is defined by the $\ell_2$-norm, with cost function $c(x, \hat{x}) = \phi(\|\Sigma^{1/2}(\hat{x} - x)\|_2) = 0.5\|\Sigma^{1/2}(\hat{x} - x)\|_2$. We focus on a diagonal cost first, and report additional experiments with off-diagonal matrices in Appendix E.2, which yield similar results. We set the first-stage cost matrix to $\Sigma_0 = \mathrm{diag}(3, 6)$, so manipulating $x_2$ is twice as costly as $x_1$, consistent with empirical evidence (Solomon, 2025; Daniel, 2024). The second-stage cost matrix is given by $\Sigma(\omega) = \mathrm{diag}(g(\omega_1), g(\omega_2)) \cdot \Sigma_0$, with $g(\omega) = \frac{1}{\omega^2}$. Smaller $\omega_i$ increase $g(\omega_i)$, raising the manipulation cost of feature $x_i$. For our decision-dependent uncertainty set, we fix $\mathbf{F}$ so $\beta$ affects only the bounds through $\mathbf{G}\beta$. Specifically, we assume the firm models its uncertainty set as $\Omega(\beta) = \{\omega \in \mathbb{R}_+^2 : \mathbf{F}\omega \leq \mathbf{h} + \mathbf{G}\beta\}$, with

$$\mathbf{F} = \begin{bmatrix} 0.9 & 0.3 \\ 0.3 & 0.6 \end{bmatrix}, \ \mathbf{h} = \begin{bmatrix} 0.5 \\ 0.5 \end{bmatrix}, \ \mathbf{G} = \begin{bmatrix} 1.5 & 0.9 \\ 0.6 & 1.5 \end{bmatrix}.$$

Larger diagonal entries of $\mathbf{G}$ indicate that each $\beta_i$ primarily governs manipulation of its own feature, while off-diagonal terms capture spillovers (e.g., extracurricular weight weakly relaxing SAT constraints). See Appendix E.1.2 for details on the rationale for the specific numerical choices.

We compare our approach against a baseline classifier that ignores the dependence of manipulation costs on decisions, which we term the decision-independent (DI) classifier. Our decision-dependent (DD) classifier, in contrast, accounts for this dependence and is obtained via our two-stage robust optimization framework with decision-dependent uncertainty costs. For the baseline in the first stage, and for both models in the second stage, we optimize the strategic hinge loss in (7) with respect to $\Sigma_0$ and realized cost matrices $\Sigma(\omega^k)$, where $\omega^k \in \Omega(\beta)$ (see Appendix E.1.3 for details on this optimization procedure). Performance is measured over 25 independent instances, each with 100 new test points sampled from the 10,000-point dataset. Second-stage results are further averaged over 10 realizations of $\omega$. We implemented the Benders C&CG algorithm in Python using GurobiPy.

Table 1 compares the performance of the DD and DI classifiers using the following metrics: (i) the actual 0-1 loss due to misclassification ("0-1 Loss" in the table), (ii) the total number of manipulations across all agents ("Manip." in the table), (iii) the number of manipulations by qualified ($y = 1$) agents ("Qual. Manip." in the table), and (iv) the number of manipulations by unqualified ($y = -1$) agents ("Unqual. Manip." in the table).

Our findings verify that the DD classifier adjusts manipulation costs in the second stage in a manner that improves accuracy relative to its DI counterpart. Specifically, as shown in Table 1, the DD classifier

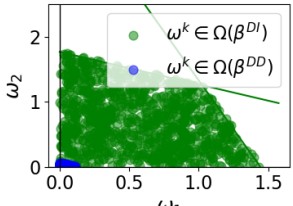

*Figure 1.* Uncertainty sets $\Omega(\beta^{\mathrm{DD}})$ vs. $\Omega(\beta^{\mathrm{DI}})$.

yields much lower second-stage loss (7.43 vs. 23.74) and manipulations (2.5 vs. 18.33) than DI. This advantage comes from anticipating how decisions affect manipulation costs, making manipulation harder ($\omega_i < 1, \forall i$). Figure 1 visualizes this: our DD classifier's uncertainty set $\Omega(\beta^{\mathrm{DD}})$ is strictly smaller than that of the baseline DI classifier $\Omega(\beta^{\mathrm{DI}})$, and induces higher manipulation costs in the second stage.

*Table 2.* Optimality gap for different approximations of the two-stage robust optimization problem (10).

| Settings | | $(t^q_{2,\max}, t^\omega_{2,\max})$ | | | | | |
|---|---|---|---|---|---|---|---|
| First-stage | Second-stage | $(0.2, 0.2)$ | $(1, 0.5)$ | $(2, 1)$ | $(0.5, 0.5)$ | Avg Gap | Avg Time |
| Approx. & Linear | Approx. & Linear | 25% | 8% | 44% | 21% | 24.5% | 11.76 s |
| Approx. & Nonlinear | Approx. & Linear | 23% | 5% | 35% | 16% | 19.75% | 13.95 s |
| Exact & Nonlinear | Approx. & Linear | 20% | 2% | 10% | 11% | 10.75% | 14.63 s |
| Exact & Nonlinear | Approx. & Nonlinear | – | – | – | – | 0.2216% | 3.11 h |
| Exact & Nonlinear | Exact & Nonlinear | – | – | – | – | 0% | 4.88 h |

In the first stage, DD performs slightly worse (loss 26.80 vs. 25.92), but overall the dependency-aware model yields clear gains: total loss drops from 49.66 to 34.23, and manipulations fall from 52.17 to 39.50. These results confirm that modeling decision-dependent costs reduces both firm error and agents' induced strategic manipulations.

**Scalability.** We have also extended our experiments to higher-dimensional synthetic data (3-, 4-, 10-, 30-, and 60-dimensional feature spaces). We observe that the maximum learning time for finding the DD-classifier under our proposed method is 132.38 seconds, supporting the scalability of our approach. Moreover, our observed findings on DD vs. DI classifiers remain consistent as dimensionality increases. Detailed results are provided in Appendix E.3.

**Optimality gap analysis.** We next assess the gap between the solution quality (in terms of the cumulative two-stage loss) obtained by our approximation, against the optimal solution of the original TSRO obtained via brute-force search. While doing so, we also consider four intermediate settings that isolate the impact of each key approximation and reformulation in both the first- and second-stage problems, as well as the choice of McCormick envelope upper bounds, on the gap. See Appendix E.4 for details of the experimental setup. The results are summarized in Table 2.

We first highlight how reducing the level of approximation in the first stage reduces the optimality gap. Starting from the fully approximated and reformulated model, which has an average optimality gap of 24.5% compared to the brute-forced solution, the gap decreases by an average of 4.75% when allowing the first-stage model to be nonlinear and only approximating the dual-norm in it, and by 13.75% once we also remove its dual-norm approximation. Now, moving onto the approximations of the second-stage (while keeping the first-stage in its original form), allowing the second-stage model to be nonlinear and only approximating the dual-norm in it, the optimality gap nearly vanishes, though this comes at a substantially higher computational cost (3 *hours* vs. 14.63 *seconds*). Notably, solving the original TSRO requires approximately 4.88 *hours* as opposed to the 11.76 *seconds* average required by our method.

We also note the impact of McCormick envelope upper bounds on the gap. While bounds of $(1, 0.5)$ yield rela-

tively small gaps (8%, 5%, and 2%), tighter bounds such as $(0.2, 0.2)$ can increase the gap (25%, 23%, and 20%), and looser bounds $(2, 1)$ also lead to larger gaps (44%, 35%, and 10%). This highlights the need for careful parameter calibration for improving the optimality gap, though the notable computational advantage of our method persists regardless.

Finally, we also compare the classifier found through our method to the optimal solution from brute-force search in terms of the uncertainty set, and the total number of manipulations by qualified and unqualified agents, emerging under each. We find that while, as expected, the optimal solution has a stricter uncertainty set and is better at curbing manipulations, the performance gap remains small, verifying that our approximation is adopting the same targets for reducing uncertainty and shaping of strategic behavior as the optimal solution. See Appendix E.5 for details.

## 7. Conclusion

We proposed a two-stage robust optimization framework for strategic classification with decision-dependent uncertainty, where future manipulation costs are shaped by past algorithmic decisions. Following this formulation, we proposed problem-specific approximations and relaxations that enabled solving this problem. Through semi-synthetic data experiments that leverage real university admissions data, we showed that the decision-dependent classifier trades slight first-stage accuracy for substantially lower overall loss and manipulation, improving robustness to strategic behavior over time. Future work directions include extending the framework to multistage problems with endogenous uncertainty, developing theoretical bounds on the relaxation gaps, and studying decision-dependent costs as functions of both endogenous (e.g., economic indicators) and exogenous (e.g., policy) contextual factors. The latter calls for new methods to learn uncertain costs and could lead to a broader framework for decision-dependent contextual stochastic optimization, an area that remains relatively underexplored. Besides, this perspective connects naturally to distributionally robust optimization, where polyhedral decision-dependent uncertainty sets may be generalized to ambiguity sets over probability distributions of uncertain parameters.

## Acknowledgments

This work was also supported in part by Cisco Research and the Jordan University of Science and Technology. Any opinions, findings, and recommendations expressed in this material are those of the authors and do not necessarily reflect the views of the sponsors.

## Impact Statement

This work studies decision-dependent uncertainty in the costs of strategic behavior within strategic classification. Our contributions are primarily theoretical. That said, the problem we consider is motivated by commonly arising real-world settings in which algorithmic decisions affect human behavior. We illustrate this through a motivating example in school admissions and note that similar considerations arise in lending and hiring. Modeling such interactions has the potential to improve robustness, reduce harmful incentives, and shape human-algorithm feedback loops in these critical contexts. Our analysis relies on stylized assumptions and does not capture all aspects of real-world behavior. Any future application should therefore be approached with caution and appropriate oversight.

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

## A. Additional Related Work

This appendix provides an extended discussion of the relationship between our two-stage robust optimization with a decision-dependent uncertainty set framework and reinforcement learning (RL)–based methods for sequential decision-making under uncertainty. Although our approach is not formulated within an RL paradigm, several conceptual connections exist, and we highlight key differences below.

The first point of contrast is in the known vs. unknown nature of the payoff/utility functions and environment dynamics. We formulate strategic classification as a two-stage robust optimization problem with a decision-dependent uncertainty set, where notably, the agent's feasible manipulation region (and specifically the effective manipulation cost) is an explicit function of the classifier's decision. This allows for deterministic worst-case guarantees rather than the simulation-based or asymptotic guarantees common in RL. As a result of this model, in our framework, the utility functions (firm and agent payoffs), and transition probabilities (mapping from first-stage decision to realized second-stage costs) are *known* (albeit stochastic, so lack of knowledge in their realizations is what causes uncertainty); in contrast, RL would be able to learn through repeated interactions had these factors been *unknown*. Given this difference in setup, RL algorithms would implement *approximate dynamic programming*, solving long-horizon problems through stochastic approximation, whereas our two-stage framework corresponds to *exact dynamic programming*, where both the classifier's decision and the agent's best response are solved through optimization. We also note that even though we employ an approximation of the agent's best response following (Rosenfeld & Rosenfeld, 2024), our formulation still yields an analytical closed-form surrogate for the agent's response without simulation or iterative learning.

A second point of comparison is in our consideration of robustness or worst-case guarantees in the solution. Robust reinforcement learning methods (e.g., robust MARL (Zhang et al., 2020), distributionally robust Markov games (Shi et al., 2025), Bayesian Stackelberg RL (Sengupta & Kambhampati, 2020), or ReBeL (Brown et al., 2020)) treat strategic behavior (with uncertainty about the environment's transition dynamics, rewards, or opponent policies) as part of a repeated Markov game and learn opponent responses through sampling, self-play, or iterative value-function approximation. Their *uncertainty* sets are *exogenous* or Bayesian and, crucially, (unlike the state transitions and reward realizations) the uncertainty sets do not depend on the learner's own decision. In contrast, our model of decision-dependent uncertainty sets adds a feedback loop on the uncertainty sets as well.

## B. Additional Interpretation of the Decision-Dependent Uncertainty (DDU) Set Model

In practice, as we explained briefly in the main paper, each component of the matrices $\mathbf{F}, \mathbf{h}$, and $\mathbf{G}$ encode economically meaningful constraints and can be specified using domain expertise, data-driven estimation, or both. The vector $\mathbf{h}$ determines the baseline upper bounds on the components of $\omega$, which determine the realization of the second-stage cost $\Sigma(\omega)$ via $g(\omega)$. For instance, in a simple illustrative case where $\Sigma(\omega) = \mathrm{diag}(2\omega_1, 5\omega_2)$ and $\omega_i \leq h_i$, choosing $h_i = 1$ could encode expected decreases in preparation costs based on expert assessment or historical data. More generally, $\mathbf{h}$ can be set using domain expertise (e.g., anticipated shifts in coaching or tutoring markets) or estimated from data on past behavioral responses.

The matrix $\mathbf{G}$ determines how the classifier's coefficients influence the feasible magnitude of manipulation incentives. Positive diagonal entries $G_{ii}$ indicate that increasing weight on feature $i$ expands the admissible range of the corresponding $\omega_i$, while negative values imply the opposite. Off-diagonal terms encode cross-feature effects—for example, how emphasizing one criterion may indirectly relax or tighten incentives to adjust another. These relationships can likewise be informed by domain experts (e.g., admissions officers' beliefs about how applicants reallocate effort) or estimated empirically from behavioral data linking past policy changes to observed manipulation patterns. In this way, $\mathbf{G}$ provides a structured mechanism for capturing decision–uncertainty interactions.

The matrix $\mathbf{F}$ imposes structural relationships among the components of $\omega$. Off-diagonal entries represent how increases in one type of preparation constrain or interact with other forms of preparation. For example, additional GPA-focused tutoring may limit resources available for extracurricular coaching, or vice versa. These couplings can be estimated from historical correlations in preparation behavior or specified by experts familiar with how various investments substitute for or complement each other.

## C. Proof of Lemma 5.1

We use Lemma 5.1 to upper bound the dual-norm term.

**Lemma 5.1.** *There exists a constant $M > 0$ such that*

$$\|\beta\|_{*,\Sigma(\omega)} \leq M\|\Sigma(\omega)^{-\frac{1}{2}}\|_\infty\|\beta\|_\infty.$$

*Proof.* From the *equivalence of norms theorem*, in finite-dimensional spaces, we know that for any two norms $\|\cdot\|_a$ and $\|\cdot\|_b$ on $\mathbb{R}^n$, there exist constants $m, M > 0$ such that:

$$m\|\beta\|_a \leq \|\beta\|_b \leq M\|\beta\|_a, \quad \forall\beta \in \mathbb{R}^n.$$

Noting that $\|\beta\|_{*,\Sigma(\omega)} = \|\Sigma(\omega)^{-\frac{1}{2}}\beta\|_*$ by definition, and invoking the equivalence of norms theorem for the $\ell_\infty$-norm, we have

$$\|\beta\|_{*,\Sigma(\omega)} \leq M\|\Sigma(\omega)^{-\frac{1}{2}}\beta\|_\infty .$$

Additionally, recall that the induced matrix $\infty$-norm (the *maximum absolute row sum norm*) of a matrix $A \in \mathbb{R}^{d\times d}$ is defined as

$$\|A\|_\infty = \max_{1\leq i\leq d}\sum_{j=1}^d |a_{ij}|.$$

Using this definition, we observe that for the identity matrix $I_{d\times d}$,

$$\|\Sigma(\omega)^{-\frac{1}{2}}\beta\|_\infty \leq \|\Sigma(\omega)^{-\frac{1}{2}}I_{d\times d}\beta\|_\infty .$$

Indeed, by expanding the norm directly,

$$\|\Sigma(\omega)^{-\frac{1}{2}}\beta\|_\infty = \max_i \left| \sum_{j=1}^d \Sigma(\omega)_{ij}^{-\frac{1}{2}}\beta_j \right|,$$

whereas

$$\|\Sigma(\omega)^{-\frac{1}{2}}I_{d\times d}\beta\|_\infty = \max_i \sum_{j=1}^d \left| \Sigma(\omega)_{ij}^{-\frac{1}{2}}\beta_j \right| .$$

Moreover, since the $\ell_\infty$-norm atisfies the *submultiplicativity* property, we have

$$\begin{aligned}\|\Sigma(\omega)^{-\frac{1}{2}}I_{d\times d}\beta\|_\infty &\leq \|\Sigma(\omega)^{-\frac{1}{2}}\|_\infty \cdot \|I_{d\times d}\beta\|_\infty \\ &= \|\Sigma(\omega)^{-\frac{1}{2}}\|_\infty \cdot \|\beta\|_\infty.\end{aligned}$$

Applying this leads to the stated bound on the matrix-induced norm. $\square$

*Example.* As a concrete example, consider the case where the primal norm adopted in the cost function is the commonly considered $\ell_2$-norm. Then, the dual norm is also an $\ell_2$-norm, and the following equivalence holds: $\|z\|_2 \leq \sqrt{d}\|z\|_\infty$ for $z \in \mathbb{R}^d$. Thus, we derive the upper bound:

$$\|\beta\|_{*,\Sigma(\omega)} \leq \sqrt{d}\ \|\Sigma(\omega)^{-\frac{1}{2}}\|_\infty\ \|\beta\|_\infty.$$

## D. C&CG-Based Solution Approach: Master Problem, Subproblem, and Algorithm

We present the master and subproblem formulations for a C&CG-based solution approach, following the implementation framework of Zeng & Wang (2022).

## D.1. Master problem

A single-level reformulation of the linear tri-model in (14) is required to define the master problem. In doing so, we adopt the three mild assumptions presented in the main paper (Section 5.3). For completeness, we provide further explanation of these assumptions here before proceeding with the derivations. Assumption (i) is required to maintain the two-stage framework. For example, if for a first-stage firm's decision $\beta^k$ the uncertainty set turns out to be empty ($\Omega(\beta^k) = \emptyset$), then the second-stage cost matrix $\Sigma(\omega)$ is not defined, and the second-stage problem becomes undefined. Consequently, the two-stage problem is trivially unbounded. Assumption (ii) ensures boundedness of the random factor $\omega$. This limits the influence of the first-stage decision on the manipulation cost in the second stage. In other words, it is not possible to drive $\omega$ to infinity, which implies, by the definition of $\Sigma(\omega)$ and our choice of $g(\omega)$, that the second-stage cost cannot be zero, and manipulation cannot be free. Finally, assumption (iii) helps detect infeasibility of (14) through its associated relaxation. By the definition of most two-stage optimization problems, a first-stage decision $\beta^k$ is feasible if the second-stage problem is feasible for all $\omega \in \Omega(\beta^k)$, and infeasible otherwise. Thus, the two-stage problem is infeasible if no feasible first-stage decision exists. Hence, (14) is infeasible if its relaxation is infeasible. Note that in (14) the second-stage problem is linear, and assumption (iii) guarantees that its dual is always feasible. This ensures the existence of a finite set of fixed extreme points and rays, which is critical for the "Benders C&CG" algorithm. Specifically, these extreme points and rays can be systematically used to iteratively refine the master problem and to generate feasibility and optimality cuts.

In this appendix, we will step-by-step derive this master problem. Specifically, this is achieved by first dualizing the linearized second-stage problem in (12) (Appendix D.1.1) to formulate the bilevel reformulation, leveraging the extreme points of the fixed feasible region. The resulting bilevel reformulation has a lower level complex disjoint bilinear program, which can be solved by enumerating on the extreme points and rays of the dual feasible region. This results in a linear bilevel reformulation (Appendix D.1.2). Moreover, this linear bilevel reformulation has lower-level linear programs for each extreme point and extreme ray (of the dual second-stage feasible set), whose KKT condition-based sets are enumerated to formulate the single-level reformulation (Appendix D.1.3). By considering a subset of these extreme points and rays, we achieve a relaxation of the large-scale single-level reformulation and a lower bound on its optimal value, that is, the "Master problem" (Appendix D.1.4).

### D.1.1. DUALIZING THE SECOND-STAGE PROBLEM

As mentioned before, we start by dualizing the second-stage problem to have the bilevel reformulation. Let $X \in \mathbb{R}^{N \times d}$, where each row is $x_i^\top$, for $i = 1, 2, \cdots, N$, and $Y \in \mathbb{Z}^N$. Given $s_1, s_2 \in \mathbb{R}^N$. For example, if both $\Sigma_0$, and $Q(g(\cdot))$ are diagonal matrices, the row sum of $\Sigma(\omega)^{-\frac{1}{2}}$ in constraint ((12).c), can be written as,

$$\sum_{j=1}^{d} Q(g(\omega))_{ij}^{-\frac{1}{2}} \cdot \Sigma_{0,ij}^{-\frac{1}{2}}.$$

This is a sum of linear functions in $\omega$, and simplifies to $g(\omega_i)^{-\frac{1}{2}} \cdot \sigma_{\Sigma_0,i}^{-\frac{1}{2}}$. Under our assumption that $g(\omega)^{-\frac{1}{2}}$ is linear in $\omega$, the row sum is therefore linear in $\omega$ as well. More generally, this row sum can be expressed compactly in vector form as

$$a_g \Sigma_0^{-\frac{1}{2}} \omega,$$

where $a_g$ encodes the coefficients induced by $g(\omega)^{-\frac{1}{2}}$. Below, we rewrite the second-stage problem in vector form, annotating each constraint with its corresponding dual variable.

$$\min \frac{\overrightarrow{1}}{N} s_2 \tag{15}$$

Subject to

$$
\begin{array}{lll}
s_2 \quad +(Y \odot X)(\beta'^+ - \beta'^-) + (u_* MY)z & \geq \vec{1}. & (\pi_0 \in \mathbb{R}^N) \\
-(\beta'^+ - \beta'^-) + \vec{1}t_2^q & \geq \vec{0}. & (\pi_1 \in \mathbb{R}^d) \\
(\beta'^+ - \beta'^-) + \vec{1}t_2^q & \geq \vec{0}. & (\pi_2 \in \mathbb{R}^d) \\
\vec{1}t_2^\omega & \geq -a_g \Sigma_0^{-1/2}\omega. & (\pi_3 \in \mathbb{R}^d) \\
\vec{1}t_2^\omega & \geq a_g \Sigma_0^{-1/2}\omega. & (\pi_4 \in \mathbb{R}^d) \\
\mathbf{B}_2(\beta'^+ - \beta'^-) & \geq \mathbf{d} - \mathbf{B}_1(\beta^+ - \beta^-) - \mathbf{E}\omega. & (\pi_5 \in \mathbb{R}^n) \\
-t_2^q & \geq -t_{2,max}^q. & (\pi_6 \in \mathbb{R}) \\
-t_2^\omega & \geq -t_{2,max}^\omega. & (\pi_7 \in \mathbb{R}) \\
-z + t_{2,max}^\omega t_2^q. & \geq 0. & (\pi_8 \in \mathbb{R}) \\
-z + t_{2,max}^q t_2^\omega & \geq 0. & (\pi_9 \in \mathbb{R}) \\
z - t_{2,max}^q t_2^\omega - t_{2,max}^\omega t_2^q & \geq -t_{2,max}^\omega t_{2,max}^q. & (\pi_{10} \in \mathbb{R}) \\
s_2, \beta'^+, \beta'^-, z, t_2^\omega, t_2^q & \geq 0.
\end{array}
$$

Here, $n$ denotes the number of constraints that characterize the feasible adjustment space of the classifier in the second stage, as outlined in Section 4. The following is the **LP** dual of the linearized second-stage problem:

$$
\max \mathbf{1}^\top \pi_0 + a_g(\Sigma_0^{-\frac{1}{2}}\omega)^\top \pi_3 - a_g(\Sigma_0^{-\frac{1}{2}}\omega)^\top \pi_4 + (d - \mathbf{B}_1(\beta^+ - \beta^-))^\top \pi_5 - (\mathbf{E}\omega)^\top \pi_5 \tag{16}
$$
$$
- t_{2,max}^q\,\pi_6 - t_{2,max}^\omega\,\pi_7 - t_{2,max}^\omega t_{2,max}^q\,\pi_{10}
$$

Subject to

$$
\begin{array}{lll}
\pi_0 & \leq & \dfrac{\vec{1}}{N}. & ((16).a) \\[2mm]
(Y \odot X)^\top \pi_0 + \mathbf{B}_2^\top \pi_5 - \pi_1 + \pi_2 & \leq & 0. & ((16).b) \\[1mm]
-(Y \odot X)^\top \pi_0 - \mathbf{B}_2^\top \pi_5 + \pi_1 - \pi_2 & \leq & 0. & ((16).c) \\[1mm]
u_* M(Y)^\top \pi_0 - \pi_8 - \pi_9 + \pi_{10} & \leq & 0. & ((16).d) \\[1mm]
\vec{1}\pi_1 + \vec{1}\pi_2 - \pi_6 + t_{2,max}^\omega \pi_8 - t_{2,max}^\omega \pi_{10} & \leq & 0. & ((16).e) \\[1mm]
\vec{1}\pi_3 + \vec{1}\pi_4 - \pi_7 + t_{2,max}^q \pi_9 - t_{2,max}^q \pi_{10} & \leq & 0. & ((16).f) \\[1mm]
\pi_0, \pi_2, \pi_1, \pi_3, \pi_4, \pi_5, \pi_6, \pi_7, \pi_8, \pi_9, \pi_{10} & \geq & 0.
\end{array}
$$

Let $\pi$ denote the set of all dual variables, with each $\pi_j$ for $j = 0, .., 10$ is in the proper dimension. Note that the dual feasible region is independent of both the uncertainty in $\omega$ and the first stage decision $(\beta^+ - \beta^-)$. Let the feasible set of the dual-second-stage problem be denoted by

$$
\Pi = \Big\{ \pi \geq 0, \text{ } \mathbf{Equations} \text{ } ((16).a)\text{-}((16).f) \Big\}.
$$

### D.1.2. BILEVEL REFORMULATION

Given the dual second-stage problem from the previous section, we now proceed by writing the bilevel reformulation of the linear tri-level problem (11).

$$
\min_{\mathcal{S}_{t1}} \frac{1}{N} \sum_{i=1}^N s_{1,i} + \max \Big\{ \mathbf{1}^\top \pi_0 + a_g(\Sigma_0^{-\frac{1}{2}}\omega)^\top \pi_3 - a_g(\Sigma_0^{-\frac{1}{2}}\omega)^\top \pi_4 + (d - \mathbf{B}_1(\beta^+ - \beta^-))^\top \pi_5 \tag{17}
$$

$$
-(\mathbf{E}\omega)^\top \pi_5 - t_{2,max}^q\,\pi_6 - t_{2,max}^\omega\,\pi_7 - t_{2,max}^\omega t_{2,max}^q\,\pi_{10} : \omega \in \Omega(\beta), \pi \in \Pi \Big\}.
$$

Note, as discussed before, this bilevel reformulation has a lower-level complex disjoint bilinear program. Zeng & Wang (2022) reformulate this as a linear bilevel reformulation by enumerating on the extreme points and rays of $\Pi$. Let $\mathcal{P}_\Pi, \mathcal{R}_\Pi$ be the set of extreme points and extreme rays of $\Pi$, respectively, with $K_p = |\mathcal{P}_\Pi|$ and $K_r = |\mathcal{R}_\Pi|$. By enumerating, we can further get a simpler but large-scale linear bilevel reformulation as follows:

$$\min \frac{1}{N} \sum_{i=1}^{N} s_{1,i} + \eta \tag{18}$$

Subject to

$$v_1 \in \mathcal{S}_{t1}, \tag{(18).a}$$

$$\left\{ \begin{aligned} &\eta \geq \mathbf{1}^\top \pi_0 + (d - \mathbf{B}_1(\beta^+ - \beta^-))^\top \pi_5 - t_{2,max}^q \pi_6 - t_{2,max}^\omega \pi_7 - t_{2,max}^\omega t_{2,max}^q \pi_{10} \\ &\quad + \max_{\omega \in \Omega(\beta)} \{ a_g (\Sigma_0^{-\frac{1}{2}} \omega)^\top \pi_3 + a_g(-\Sigma_0^{-\frac{1}{2}} \omega)^\top \pi_4 + (-\mathbf{E}\omega)^\top \pi_5 \} : \omega \in \Omega(\beta), \pi \in \Pi \} \end{aligned} \right\} \forall \pi \in \mathcal{P}_\Pi, \tag{(18).b}$$

$$\left\{ \begin{aligned} &\mathbf{1}^\top \gamma_0 + (d - \mathbf{B}_1(\beta^+ - \beta^-))^\top \gamma_5 - t_{2,max}^{\tilde{\mathbf{v}}} \gamma_6 - t_{2,max}^{\tilde{\mathbf{v}}} \gamma_7 - t_{2,max}^{\tilde{\mathbf{v}}} t_{2,max}^q \gamma_{10} \\ &\quad + \max_{\tilde{\mathbf{v}} \in \Omega(\beta)} \{ a_g (\Sigma_0^{-\frac{1}{2}} \tilde{\mathbf{v}})^\top \gamma_3 + a_g(-\Sigma_0^{-\frac{1}{2}} \tilde{\mathbf{v}})^\top \gamma_4 + (-\mathbf{E}\tilde{\mathbf{v}})^\top \gamma_5 \} \leq 0 \end{aligned} \right\} \forall \gamma \in \mathcal{R}_\Pi. \tag{(18).c}$$

Note that the variable $\tilde{\mathbf{v}}$ is an alias of $\omega$. Given the lower-level LPs appears in ((18).b) and ((18).c), let $\mathbf{LP}(\beta, U)$ : $\max\{ a_g (\Sigma_0^{-\frac{1}{2}} \omega)^\top U_3 + a_g(-\Sigma_0^{-\frac{1}{2}} \omega)^\top U_4 + (-\mathbf{E}\omega)^\top U_5 : \omega \in \Omega(\beta) \}$. Using the KKT conditions let $\mathcal{O}\Omega(\beta, \pi^k)$ denotes the optimal solution set of $\mathbf{LP}(\beta, \pi^k)$. Then,

$$\mathcal{O}\Omega(\beta, \pi^k) = \left\{ \begin{aligned} &\mathbf{F}(\beta)\, \omega^k \leq \mathbf{h} + \mathbf{G}(\beta^+ - \beta^-), \\ &\mathbf{F}(\beta)^\top \lambda^k \geq + a_g \Sigma_0^{-\frac{1}{2}} \pi_3^k - a_g \Sigma_0^{-\frac{1}{2}} \pi_4^k - \mathbf{E}\pi_5^k \\ &\lambda^k \odot \left( \mathbf{h} + \mathbf{G}(\beta^+ - \beta^-) - \mathbf{F}(\beta)\, \omega^k \right) = 0, \\ &\omega^k \odot \left( \mathbf{F}(\beta)^\top \lambda^k - a_g \Sigma_0^{-\frac{1}{2}} \pi_3^k + a_g \Sigma_0^{-\frac{1}{2}} \pi_4^k + \mathbf{E}^\top \pi_5^k \right) = 0 \\ &\omega^k,\, \lambda^k \geq 0 \end{aligned} \right\}, \tag{19}$$

where $\lambda^k$ represents the dual variable of constraints $\Omega(\beta)$. Similarly, we can define $\mathcal{O}\mathcal{V}(\beta, \gamma^l)$ to be the set of optimal solution for $\mathbf{LP}(\beta, \gamma^l)$ using KKT conditions.

### D.1.3. SINGLE LEVEL REFORMULATION

The bilevel reformulation in the previous section (Appendix D.1.2) is also equivalent to the single-level optimization problem in this section. Using the sets $\mathcal{O}\Omega$ and $\mathcal{O}\mathcal{V}$ defined in the previous section, we can write a single-level reformulation,

$$\min \frac{1}{N} \sum_{i=1}^{N} s_{1,i} + \eta \tag{20}$$

Subject to

$$v_1 \in \mathcal{S}_{t1},$$

$$\left\{ \eta \geq \mathbf{1}^\top \pi_0^k + (d - \mathbf{B}_1(\beta^+ - \beta^-))^\top \pi_5^k - t_{2,max}^q \, \pi_6^k - t_{2,max}^{\omega^k} \, \pi_7^k - t_{2,max}^{\omega^k} t_{2,max}^q \, \pi_{10}^k \right.$$

$$\left. + a_g(\Sigma_0^{-\frac{1}{2}}\omega^k)^\top \pi_3^k - a_g(\Sigma_0^{-\frac{1}{2}}\omega^k)^\top \pi_4^k - (\mathbf{E}\omega)^\top \pi_5^k \right\}, \; k = 1, \cdots, K_p,$$

$$(\omega^k, \lambda^k) \in \mathcal{O}\Omega(\beta, \pi^k), \; k = 1, \cdots, K_p$$

$$\left\{ \mathbf{1}^\top \gamma_0^l + (d - \mathbf{B}_1(\beta^+ - \beta^-))^\top \gamma_5^l - t_{2,max}^q \, \gamma_6^l - t_{2,max}^{\tilde{\mathbf{v}}^l} \, \gamma_7^l - t_{2,max}^{\tilde{\mathbf{v}}^l} t_{2,max}^q \, \gamma_{10}^l \right.$$

$$\left. + a_g(\Sigma_0^{-\frac{1}{2}}\tilde{\mathbf{v}}^l)^\top \gamma_3^l - a_g(\Sigma_0^{-\frac{1}{2}}\tilde{\mathbf{v}}^l)^\top \gamma_4^l - (\mathbf{E}\tilde{\mathbf{v}^l})^\top \gamma_5^l \right\} \leq 0 \right\}, \; l = 1, \cdots, K_r$$

$$(\tilde{\mathbf{v}}^l, \xi^l) \in \mathcal{O}\mathcal{V}(\beta, \gamma_1^l), \; l = 1, \cdots, K_r$$

### D.1.4. THE MASTER PROBLEM

As mentioned before, the single-level reformulation includes the use of all extreme points and rays of the dual-second-stage feasible region ($\Pi$). Hence, for a subset of these extreme points and the extreme rays sets, we have a relaxation of the single-level reformulation (($20$)) which is the "Master problem". Let $\hat{\mathcal{P}}_\Pi \subseteq \mathcal{P}_\Pi$, and $\hat{\mathcal{R}}_\Pi \subseteq \mathcal{R}_\Pi$ be the subset of the extreme points and rays, receptively. We can lower bound the problems in the following **"Master problem"**:

$$\min \frac{1}{N} \sum_{i=1}^N s_{1,i} + \eta \tag{21}$$

Subject to

$$v_1 \in \mathcal{S}_{t1}, \tag{(21).a}$$

$$\left\{ \eta \geq \mathbf{1}^\top \pi_0 + (d - \mathbf{B}_1(\beta^+ - \beta^-))^\top \pi_5 - t_{2,max}^q \, \pi_6 - t_{2,max}^\omega \, \pi_7 - t_{2,max}^\omega t_{2,max}^q \, \pi_{10} \right. \tag{(21).b}$$

$$\left. + a_g(\Sigma_0^{-\frac{1}{2}}\omega^\pi)^\top \pi_3 - a_g(\Sigma_0^{-\frac{1}{2}}\omega^\pi)^\top \pi_4 - (\mathbf{E}\omega^\pi)^\top \pi_5 \right\} \quad \forall \pi \in \hat{\mathcal{P}}_\Pi,$$

$$(\omega^\pi, \lambda^\pi) \in \mathcal{O}\Omega(\beta, \pi^\pi), \; \forall \pi \in \hat{\mathcal{P}}_\Pi \tag{(21).c}$$

$$\left\{ \mathbf{1}^\top \gamma_0 + (d - \mathbf{B}_1(\beta^+ - \beta^-))^\top \gamma_5 - t_{2,max}^q \, \gamma_6 - t_{2,max}^{\tilde{\mathbf{v}}} \, \gamma_7 - t_{2,max}^{\tilde{\mathbf{v}}} t_{2,max}^q \, \gamma_{10} \right. \tag{(21).d}$$

$$\left. + a_g(\Sigma_0^{-\frac{1}{2}}\tilde{\mathbf{v}}^\gamma)^\top \gamma_3 - a_g(\Sigma_0^{-\frac{1}{2}}\tilde{\mathbf{v}}^\gamma)^\top \gamma_4 - (\mathbf{E}\tilde{\mathbf{v}}^\gamma)^\top \gamma_5 \right\} \leq 0 \quad \forall \gamma \in \hat{\mathcal{R}}_\Pi.$$

$$(\tilde{\mathbf{v}}^\gamma, \xi^\gamma) \in \mathcal{O}\mathcal{V}(\beta, \gamma_1^\gamma), \; \forall \gamma \in \hat{\mathcal{R}}_\Pi \tag{(21).e}$$

### D.2. Subproblems

**Subproblem 1 (SP1)** The first subproblem (SP1) is formulated to check the feasibility of the current first-stage $\beta^*$, which by definition is feasible if the second-stage problem is feasible to all scenarios in the uncertainty set $\Omega(\beta^*)$.

$$\eta_f(\beta^*) = \max_{\omega \in \Omega(\beta^*)} \min \mathbf{1}^\top \tilde{\mathbf{S}}_0 + \mathbf{1}^\top \tilde{\mathbf{S}}_1 + \mathbf{1}^\top \tilde{\mathbf{S}}_2 + \mathbf{1}^\top \tilde{\mathbf{S}}_3 + \mathbf{1}^\top \tilde{\mathbf{S}}_4 + \mathbf{1}^\top \tilde{\mathbf{S}}_5 + \tilde{\mathbf{S}}_6 + \tilde{\mathbf{S}}_7 \tag{22}$$

$$+ \tilde{\mathbf{S}}_8 + \tilde{\mathbf{S}}_9 + \tilde{\mathbf{S}}_{10}$$

S.t.

$$
\begin{array}{rcl}
\tilde{\mathbf{S}}_0 + s_{2,i} + y_i\left((\beta'^+ - \beta'^-)^\top x_i + u_* M z\right) & \geq & 1, \quad i = 1, \ldots, N, \\[6pt]
\tilde{\mathbf{S}}_1 - (\beta'^+ - \beta'^-)_j & \geq & -t_2^q, \quad j = 1, \ldots, d, \\[6pt]
\tilde{\mathbf{S}}_2 + (\beta'^+ - \beta'^-)_j & \geq & -t_2^q, \quad j = 1, \ldots, d, \\[6pt]
\tilde{\mathbf{S}}_3 - \sum_j [\Sigma(\omega)^{-\frac{1}{2}}]_{ij} & \geq & -t_2^\omega, \quad j = 1, \ldots, d, \\[6pt]
\tilde{\mathbf{S}}_4 + \sum_j [\Sigma(\omega)^{-\frac{1}{2}}]_{ij} - t_2^\omega & \geq & -t_2^\omega, \quad j = 1, \ldots, d, \\[6pt]
\tilde{\mathbf{S}}_5 + \mathbf{B}_2(\beta'^+ - \beta'^-) & \geq & \mathbf{d} - \mathbf{B}_1\beta^* - \mathbf{E}\omega, \\[6pt]
\tilde{\mathbf{S}}_6 - t_2^q & \geq - & t_{2,max}^q \\[6pt]
\tilde{\mathbf{S}}_7 - t_2^\omega & \geq & -t_{2,max}^\omega \\[6pt]
\tilde{\mathbf{S}}_8 - z & \geq & -t_{2,max}^\omega t_2^q \\[6pt]
\tilde{\mathbf{S}}_9 - z & \geq & -t_{2,max}^q t_2^\omega \\[6pt]
\tilde{\mathbf{S}}_{10} + z & \geq & t_{2,max}^q t_2^\omega + t_{2,max}^\omega t_2^q - t_{2,max}^\omega t_{2,max}^q \\[6pt]
\tilde{\mathbf{S}}_j, z, s_{2,i}, \beta'^+, \beta'^-, t_2^q, t_2^\omega & \geq & 0, \text{ for } i = 1, 2, \ldots, N, \text{ for } j = 1, 2, \ldots, 12.
\end{array}
$$

Accordingly, the linearized tri-model in Equation (11) and all its equivalences are feasible for $\beta^*$ if and only if **SP1** objective value is zero ($\eta_f(\beta^*) = 0$).

**[Case A]:** When $\eta_f(\beta^*) = 0$, meaning that the second-stage problem is feasible for all $\omega \in \Omega(\beta^*)$, We then solve the second subproblem (SP2) to assess the worst-case performance of $\beta^*$ by identifying the worst-case realization $\omega_s^*$ and its corresponding second-stage cost $\eta_s(\beta^*)$.

$$
\textbf{(SP2)} \quad \eta_s(\beta^*) = \max_{\omega \in \Omega(\beta^*)} \min \left\{ \frac{1}{N} \sum_{i=1}^N s_{2,i} : \text{Equation ((12).a)-((12).h))} \right\} \tag{23}
$$

**(SP2)** can be addressed by reformulating the minimization problem via its KKT conditions or equivalently through its dual problem. In both computational approaches, the optimal dual variables, denoted by $\pi^*$, correspond to an extreme point of $\Pi$. From the lower-level linear program in the bilevel reformulation, denoted as $\mathbf{LP}(\beta, U)$, it follows that

$$
\begin{aligned}
\eta_s(\beta^*) = \mathbf{1}^\top \pi_0^* + (d - \mathbf{B}_1(\beta^{+*} - \beta^{-*}))^\top \pi_5^* - t_{2,max}^q \, \pi_6^* - t_{2,max}^\omega \, \pi_7^* - t_{2,max}^\omega t_{2,max}^q \, \pi_{10}^* \\
+ \mathrm{LP}(\beta^*, \pi^*).
\end{aligned} \tag{24}
$$

**[Case B]:** Conversely, if $\eta_f(\beta^*) > 0$, the optimal solution to (22), denoted by $\omega_f^*$, renders the second-stage problem infeasible. In this situation, we solve the third subproblem **(SP3)**, which corresponds to the dual of the second-stage problem evaluated at $\omega_f^*$.

$$
\begin{aligned}
\textbf{(SP3)} \quad \max \Big\{ &\mathbf{1}^\top \pi_0 + (d - \mathbf{B}_1(\beta^{+*} - \beta^{-*}))^\top \pi_5 - t_{2,max}^q \, \pi_6 - t_{2,max}^\omega \, \pi_7 - t_{2,max}^\omega t_{2,max}^q \, \pi_{10} \\
&+ a_g(\Sigma_0^{-\frac{1}{2}} \omega_f^*)^\top \pi_3 - a_g(\Sigma_0^{-\frac{1}{2}} \omega_f^*)^\top \pi_4 - (\mathbf{E}\omega_f^*)^\top \pi_5 : \pi \in \Pi \Big\}
\end{aligned} \tag{25}
$$

Note that **(SP3)** is unbounded with respect to $(\beta^*, \omega_f^*)$, its solution identifies an extreme ray of $\Pi$, denoted by $\gamma^*$, along which the objective value diverges to infinity. By convention, the corresponding worst-case second-stage cost $\eta_s(\beta^*)$ is set to $+\infty$.

## D.3. Algorithm

When the two-stage robust optimization (2-Stg RO) problem has a decision-independent uncertainty (DIU) set, classical Column-and-Constraint Generation (C&CG) iteratively solves a master problem by adding one recourse problem per identified worst-case scenario. As shown in Zeng & Wang (2022), for 2-Stg RO with decision-dependent uncertainty (DDU) and a single-level reformulation (Appendix D.1.3), this strategy can be extended using a parametric framework.

The resulting **Benders C&CG algorithm** dynamically generates worst-case scenarios and the corresponding dual-based optimality or feasibility cuts, refining the master problem over iterations. Below are the algorithmic steps:

1. **Initialization**: Set lower bound LB $= -\infty$, upper bound UB $= +\infty$, iteration index $k = 1$, cut sets $\hat{\mathcal{P}}_\Pi, \hat{\mathcal{R}}_\Pi = \emptyset$, and choose a convergence tolerance $\epsilon > 0$.

2. **Master Problem (MP)**: Solve the master problem in (21) (Appendix D.1.4) to obtain candidate solution $(v_1^k, \eta^k)$. Set LB $= \frac{1}{N} \sum_{i=1}^N s_{1,i}^k + \eta^k$.

3. **Subproblem 1 (SP1)**: For given $\beta^k$, solve subproblem (SP1) in (22) (Appendix D.2) to compute $\eta_f(\beta^k)$ and corresponding scenario $\omega_f^k$.

4. **Cut Generation**:
   - **(Case A)**: If $\eta_f(\beta^k) = 0$, solve subproblem (SP2) in (23) to obtain $\eta_s(\beta^k)$, scenario $\omega_s^k$, optimal dual solution $\pi^k$. Update $\hat{\mathcal{P}}_\Pi \leftarrow \hat{\mathcal{P}}_\Pi \cup \{\pi^k\}$. Add optimality cuts from ((21).b)–((21).c).
   - **(Case B)**: If $\eta_f(\beta^k) > 0$, solve subproblem (SP3) in (25) to obtain extreme ray $\gamma^k$, and set $\eta_s(\beta^k) = +\infty$. Update $\hat{\mathcal{R}}_\Pi \leftarrow \hat{\mathcal{R}}_\Pi \cup \{\gamma^k\}$. Add feasibility cuts from ((21).d)–((21).e).

5. **Upper Bound Update**: Set $\mathrm{UB}^k = \frac{1}{N} \sum_{i=1}^N s_{1,i}^k + \eta_s(\beta^k)$, and update UB $= \min\{\mathrm{UB}, \mathrm{UB}^k\}$.

6. **Convergence Check**: If $\mathrm{UB} - \mathrm{LB} \le \epsilon$, terminate and return $\beta^k$ as the optimal first-stage solution. Otherwise, increment $k \leftarrow k + 1$ and return to Step 2.

# E. Implementation, Experimental Details, and Additional Numerical Experiments

## E.1. Implementation and experimental details

### E.1.1. UNIVERSITY ADMISSION DATA GENERATION

We use two features: $x_1$ (SAT score) and $x_2$ (number of extracurricular activities). Here, we provide additional details on how these features are generated. For the SAT feature ($x_1$), we rely on the nationwide score distribution reported in the SAT Suite of Assessments Annual Report (College Board, 2023), which provides participation statistics, score intervals (e.g., 1400–1600, 1200–1390), and overall summary statistics (mean and standard deviation). This empirical distribution forms the basis for generating SAT scores in our experiment (Figure 2a). For the extracurricular activities feature ($x_2$), we draw on data from (Park et al., 2025), which analyzes student-reported counts of activities from the Common Application ("Common App") database (Figure 2b). To maintain temporal consistency, we align SAT data with the years examined in (Park et al., 2025) (2018 and 2019). Finally, both features are standardized using $z$-score normalization—subtracting the mean and dividing by the standard deviation—so that each has mean zero and unit variance. The resulting dataset consists of two-dimensional features $x \in \mathbb{R}^2$, with each component sampled from its respective empirical distribution.

Admissions decision-making is informed by empirical weightings from (National Association for College Admission Counseling, 2025), which reports that academic factors such as GPA, curriculum rigor, and standardized test scores carry substantially more weight than extracurricular activities in undergraduate admissions. On average, $45.17\%$ of colleges report attributing considerable importance to SAT scores, compared to $6.83\%$ for extracurricular activities. Normalizing these proportions to relative weights yields approximately $0.85$ for SAT scores and $0.15$ for extracurricular activities. Labels are then assigned according to the following linear decision boundary defined by these weights.

$$y = \begin{cases} 1 & \text{if } 0.85x_1 + 0.15x_2 > \theta, \\ -1 & \text{otherwise,} \end{cases}$$

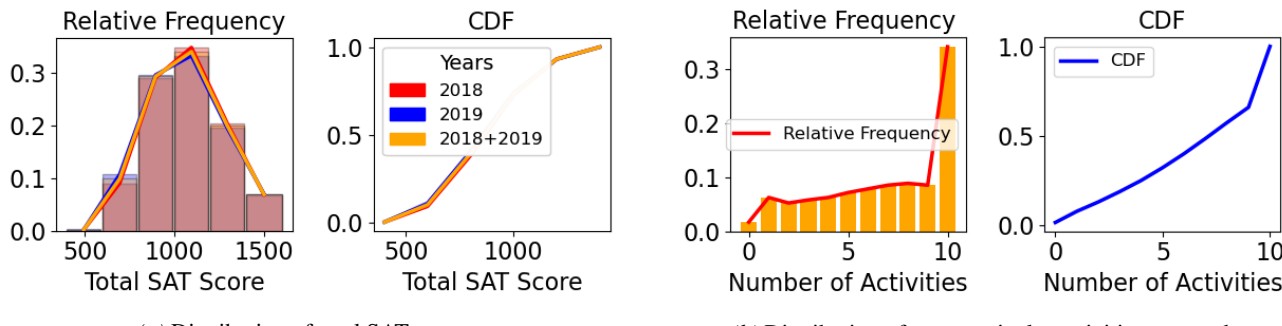

*(a)* Distribution of total SAT scores.   *(b)* Distribution of extracurricular activities reported.

*Figure 2.* University admission features distribution

Here, $\theta$ represents a plausible qualification threshold. In practice, most highly selective universities admit students with SAT scores near the 75th percentile or higher (Crimson Education, 2025). For extracurricular activities (ECs), we adopt a benchmark of 4–5 activities. This choice is motivated by evidence that the Common App allows up to 8–10 activities, and recent work suggests that lowering this maximum may help mitigate equity concerns (Park et al., 2025). Accordingly, we define the qualification threshold $\theta$ as a composite of a 75th-percentile SAT score (1210) and a moderate extracurricular profile of 4–5 activities. This provides a principled proxy for admissions decisions while ensuring that the classification task remains both realistic and non-trivial. To simulate noisy environments, we adopt *feature-dependent noise* (Zhang et al., 2021; Smart & Carneiro, 2023), in which labels are flipped with probability $\mathbb{P}_{\text{noise}}(x)$ proportional to a point's proximity to the decision boundary. In this formulation, samples closer to the threshold are more likely to be mislabeled, reflecting the ambiguity inherent in real admissions decisions. The final dataset consists of 10,000 points.

$$\mathbb{P}_{\text{noise}}(x) = 0.5 \exp\left(-\left(\tfrac{0.85x_1 + 0.15x_2}{0.1}\right)^2\right).$$

### E.1.2. COST AND UNCERTAINTY DETAILS

**First-stage cost:** We fix the cost matrix as $\Sigma_0 = \text{diag}(3, 6)$, which implies that manipulation of the extracurricular activities feature ($x_2$) is twice as expensive as manipulation of the SAT feature ($x_1$). This assumption is consistent with empirical evidence. Sustained extracurricular participation (e.g., sports) often incurs annual costs of \$700–\$1,500 for a single primary activity (Solomon, 2025). Because students typically engage in such activities across multiple years of high school (3–4 years) to demonstrate continuity and leadership, families may spend \$2,000–\$6,000 or more on one activity alone. By contrast, SAT preparation incurs lower or moderate costs, typically ranging from \$100–\$1,400 (Daniel, 2024), with free or low-cost options also available. Moreover, SAT preparation expenses are generally concentrated within a single year rather than sustained throughout high school.

**More explanation on decision-dependent uncertainty.** To construct the decision-dependent uncertainty set, we fix the interaction matrix $\mathbf{F}$ to be constant and independent of the first-stage decision $\beta$. This design ensures that $\beta$ influences only the *bounds* of the uncertainty set, through the term $\mathbf{G}\beta$, rather than altering the relative interaction between $\omega_1$ and $\omega_2$. Given,

$$\mathbf{F} = \begin{bmatrix} 0.9 & 0.3 \\ 0.3 & 0.6 \end{bmatrix}, \quad \mathbf{h} = \begin{bmatrix} 0.5 \\ 0.5 \end{bmatrix}, \quad \mathbf{G} = \begin{bmatrix} 1.5 & 0.9 \\ 0.6 & 1.5 \end{bmatrix}.$$

The diagonal entries of $\mathbf{G}$ are set larger than the off-diagonal entries, reflecting that $\beta_i$ primarily governs manipulation costs for its corresponding feature $x_i$, rather than for $x_j$ with $i \neq j$. The baseline vector $\mathbf{h} = (0.5, 0.5)^\top$ captures general access to resources (e.g., widespread availability of SAT preparation materials and extracurricular opportunities). For instance, the inequality $0.9\omega_1 + 0.3\omega_2 \leq 0.5 + 1.5\beta_1 + 0.9\beta_2$ demonstrates how policy weights shape the relative ease of manipulation across features. Placing greater weight on SAT scores ($\beta_1$) makes SAT manipulation ($\omega_1$) relatively easier, as students shift resources toward SAT preparation, which is typically cheaper at the margin than sustained extracurricular participation. However, manipulation is not fully independent across features: the cross-effect coefficient (0.9) indicates that increasing $\beta_2$ also relaxes the SAT constraint, though less strongly. Conversely, raising the weight on extracurriculars ($\beta_2$) reduces the cost of manipulating $\omega_2$, with some spillover from SAT emphasis captured by the cross-effect coefficient (0.6). These coupled constraints emphasize that admissions criteria are not isolated levers: increasing emphasis on one dimension necessarily

reshapes incentives in the other. In practice, this captures the substitutability of effort, as students reallocate limited time and financial resources between SAT preparation and extracurricular involvement in response to admissions signals. The model thus reflects both direct responsiveness (own-feature effects) and cross-substitution (spillovers), making explicit the interdependence between academic and non-academic factors in holistic admissions.

### E.1.3. OPTIMIZATION PROCEDURES FOR DI BASELINE AND SECOND-STAGE CLASSIFIERS

The first-stage decision-independent baseline classifier, $\beta^{\text{DI}}$, is trained via mini-batch gradient descent (batch size 100) to minimize the strategic hinge loss in (7) under the fixed cost matrix $\Sigma_0$. Convexity is essential for applying gradient descent; therefore, we include a regularization term of the form $R_{\Sigma_0}(\beta) + \lambda_{reg} u_* \|\beta\|_{*,\Sigma_0}$. By Proposition 4.3 of Rosenfeld & Rosenfeld (2024), adding this regularizer and selecting $\lambda_{\text{reg}} \geq \mathbb{P}_{PXY}(Y = 1)$ guarantees convexity. Note also that the second-stage optimal classifiers for both models are obtained in the same manner but with respect to realized cost matrices $\Sigma(\omega^k)$, where $\omega^k \in \Omega(\beta)$.

### E.1.4. COMPUTATIONAL SETUP

All experiments were conducted on a personal laptop (HP OMEN Slim 16) with an Intel Core Ultra 7 255H CPU and 32 GB RAM, running Windows 11 (64-bit). No GPU acceleration was used. The implementation was in Python 3.8.3 with Gurobi 12.0.1. Our implementation was written in Python 3.8.3 and used Gurobi 12.0.1 (via GurobiPy).

### E.2. Additional numerical experiments with an off-diagonal cost matrix

In this section, we conduct two experiments to generalize the cost matrices considered in the simulations, both including off-diagonal elements: one with positive correlations and another with negative correlations between features. We set the first-stage cost matrix to

$$\Sigma_0 = \begin{bmatrix} 3 & 1 \\ 1 & 6 \end{bmatrix}.$$

The diagonal entries represent the marginal cost of moving each feature independently. If $v = x' - x = (v_1, v_2)^\top$, then

$$v^\top \Sigma_0 v = 3v_1^2 + 2v_1 v_2 + 6v_2^2.$$

Ignoring the cross term for a moment, a unit change in $x_2$ contributes 6 units of quadratic cost, whereas a unit change in $x_1$ contributes only 3; manipulating $x_2$ is therefore roughly *twice as costly* as manipulating $x_1$. The off-diagonal entries capture *interaction* between coordinates. The term $2v_1 v_2$ implies that the cost of moving both features simultaneously is not merely the sum of their marginal cost. If $2v_1 v_2 > 0$ (same sign), the total cost increases; coordinated increases are penalized more heavily. However, if $v_1$ and $v_2$ have opposite signs, then the cross term becomes negative and partially offsets the marginal costs, making some compensating movements cheaper.

Geometrically, the manipulation costs depend on both the magnitude and the *direction* of the change. In the second stage, the cost matrix is a function of the uncertainty vector $\omega = (\omega_1, \omega_2)^\top$ via the function $g(\omega) = \begin{bmatrix} g_1(\omega_1) \\ g_2(\omega_2) \end{bmatrix}$, which is component-wise separable, similar to the n the diagonal-cost experiments, we set $g_i(\omega_i) = \frac{1}{\omega_i^2}$. Thus, each uncertainty component $\omega_i$ affects only the $i$th entry of $g(\omega)$. Let the eigen-decomposition of $\Sigma_0$ given by:

$$\Sigma_0 = V \Lambda V^\top, \qquad \Lambda = \text{diag}(\lambda_1, \lambda_2),$$

where $V$ is orthogonal and $\lambda_1, \lambda_2 > 0$. The off-diagonal second-stage cost matrix is given by

$$\Sigma(\omega) = Q(g(\omega)) \cdot \Sigma_0, \qquad Q(g(\omega)) := V \, \text{diag}\big(g(\omega)\big) V^\top \Sigma_0^{-1}.$$

**Positive correlation.** For the positive-correlation case, we use the same decision-dependent uncertainty set as in the diagonal-cost experiments. All entries of **G** are positive: the diagonal terms indicate how each coefficient $\beta_i$ expands its own uncertainty range, while the off-diagonal terms represent cross-feature spillovers (e.g., increasing weight on extracurriculars slightly enlarges SAT-manipulation uncertainty). Table 3 shows that our analysis continues to hold under an off-diagonal cost matrix with positive correlation.

Relative to the diagonal-cost case, we observe substantially greater average manipulation by unqualified agents during both stages when the cost matrix contains off-diagonal elements, under both the DD and DI classifiers (13.44 vs. 19.04, 24.64 vs. 21.36, 2.30 vs. 10.07, and 16.46 vs. 25.67). This increased manipulation leads to larger total losses for the classifiers as well (34.23 vs. 44.09 and 49.66 vs. 60.55). The reason is that the off-diagonal matrix alters the structure of manipulation costs: rather than penalizing each feature independently, the cost now depends on how features are adjusted jointly, and the off-diagonal entries generate directions in feature space that are effectively cheaper to move along. Nonetheless, even this increase in second-stage manipulation by unqualified agents remains lower than that observed in the first stage for the DD classifier (10.07 vs. 19.04).

*Table 3.* Off-diagonal cost matrix with positive correlation.

| Metric | First-stage DD | First-stage DI | Second-stage DD | Second-stage DI | Total DD | Total DI |
|---|---|---|---|---|---|---|
| 0–1 Loss | 28.04±0.79 | 25.40±0.75 | 16.05±1.42 | 35.15±3.54 | **44.09** | 60.55 |
| Manipulations | 39.72±0.88 | 32.68±0.91 | 10.44±1.47 | 28.62±3.59 | **50.16** | 61.30 |
| Qualified Manip. | 20.68±0.81 | 11.20±0.55 | 0.35±0.08 | 2.94±0.80 | **21.03** | 14.14 |
| Unqualified Manip. | 19.04±0.64 | 21.36±0.86 | 10.07±1.39 | 25.67±3.60 | **29.11** | 47.03 |

**Negative correlation.** We also consider an uncertainty set with

$$\mathbf{F} = \begin{bmatrix} 0.9 & 0.3 \\ 0.3 & 0.6 \end{bmatrix}, \quad \mathbf{h} = [5, 5], \quad \mathbf{G} = \begin{bmatrix} -1.5 & -0.9 \\ -0.6 & -1.5 \end{bmatrix}.$$

Here, the negative entries of $\mathbf{G}$ imply that increasing a coefficient $\beta_i$ tightens the corresponding uncertainty bound; that is, placing more weight on a feature reduces the feasible range of manipulation for that component. This captures settings in which emphasizing a feature discourages strategic adjustment of the associated attribute. Table 4 shows that our analysis continues to hold under an off-diagonal cost matrix with negative correlation.

As before, we observe substantially larger second-stage manipulation by unqualified agents and higher total losses relative to the diagonal-cost case (16.12 vs. 13.44, 2.30 vs. 8.15, and 16.46 vs. 27.90), which results in higher total losses for both classifiers (40.44 vs. 34.23 and 67.11 vs. 49.66). Although there is a slight decrease under the DI classifier in the first stage, the DD classifier continues to achieve lower total losses.

As in the diagonal-cost experiments, the reductions observed under both off-diagonal models arise because the DD classifier anticipates the effect of its decisions, making manipulation costlier and thus more difficult. Figures 3a and 3b illustrate this: $\Omega(\beta^{\mathrm{DI}})$ contains substantially larger values of $\omega$ than $\Omega(\beta^{\mathrm{DD}})$, reflecting cheaper manipulation under DI.

*Table 4.* Off-diagonal cost matrix with negative correlation.

| Metric | First-stage DD | First-stage DI | Second-stage DD | Second-stage DI | Total DD | Total DI |
|---|---|---|---|---|---|---|
| 0–1 Loss | 26.36±1.01 | 24.44±0.93 | 14.08±0.85 | 42.67±4.66 | **40.44** | 67.11 |
| Manipulations | 27.08±1.07 | 29.08±0.85 | 8.64±0.84 | 31.81±5.96 | **35.72** | 60.89 |
| Qualified Manip. | 10.96±0.66 | 9.32±0.51 | 0.46±0.08 | 3.89±0.64 | **11.42** | 13.21 |
| Unqualified Manip. | 16.12±0.72 | 19.64±0.75 | 8.15±0.83 | 27.90±6.29 | **24.27** | 47.54 |

### E.3. Scalability to higher-dimensional feature spaces

For the scalability of our method, specifically, the number of feature dimensions affects the number of extreme points in the feasible set of the dual second-stage problem. In the worst case, the iteration complexity can grow exponentially in the number of extreme points of this feasible set. That said, convergence time in practice is often low. We extend the experiment to 3-, 4-, 10-, 30-, and 60-dimensional feature spaces. Table 5 reports the computation time across dimensions, with a maximum training time of 132.38 seconds at 60-dimensional settings.

In particular, combining cuts generated via Benders decomposition with C&CG leads to faster convergence, requiring fewer iterations than pure Benders decomposition. This is because, unlike Benders decomposition, which adds only constraints, the C&CG component introduces both new columns (recourse decisions) and constraints (subproblem scenarios).

Due to the high computational cost of evaluating true strategic behavior in higher dimensions, we report full experimental

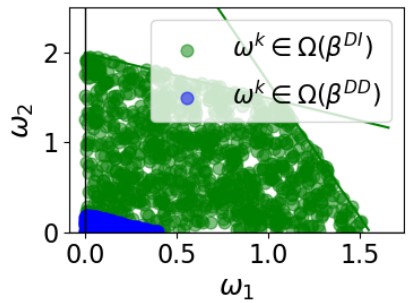
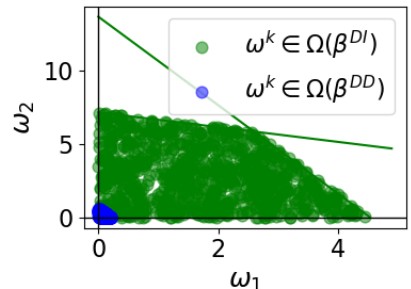

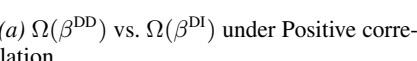

*(a)* $\Omega(\beta^{\mathrm{DD}})$ vs. $\Omega(\beta^{\mathrm{DI}})$ under Positive correlation.

*(b)* $\Omega(\beta^{\mathrm{DD}})$ vs. $\Omega(\beta^{\mathrm{DI}})$ under Negative correlation.

*Figure 3.* Second-stage decision-dependent uncertainty sets under off-diagonal cost matrices.

*Table 5.* Computation time of the proposed two-stage robust optimization model with decision-dependent uncertainty across feature-space dimensions (3-60).

| Feature dimensions | 2 | 3 | 4 | 10 | 30 | 60 |
|---|---|---|---|---|---|---|
| Time (seconds) | 12.95 | 15.6 | 16.1 | 30 | 94.45 | 132.38 |

results only for the 10-dimensional settings. In contrast, training remains efficient even in higher dimensions. Table 6 shows that our findings remain consistent as dimensionality increases.

**Experimental setup for the 10-dimensional settings.** The first two dimension are kept the same as in the two-dimensional experiment, while for $x_3, \cdots, x_{10} \sim Uniform(0, 10)$. Similarly, Labels are assigned as $y = 1$ if $\theta^\top x > 0$ and $y = -1$ otherwise, where $\theta = [0.102, -0.347, 0.251, 0.314, -0.652, -0.435, 0.043, -0.106, -0.006, -0.285]^\top$.

The students in the first stage can manipulate under the following cost matrix

$$\Sigma_0 = \mathrm{diag}\Big(5.15,\ 6.52,\ 4.58,\ 4.52,\ 7.95, 6.20,\ 5.31,\ 4.59,\ 3.49,\ 5.29\Big).$$

We are following the same modeling of the second-stage cost matrix as in the two-dimensional settings using the same $g(\omega) = \frac{1}{\omega^2}$, and $\Sigma(\omega) = \mathrm{diag}(g(\omega)_1, \cdots, g(\omega)_{10})$. The decision-dependent uncertainty set again fixes $\mathbf{F}$ so $\beta$ affects only the bounds through $\mathbf{G}\beta$. We assume the firm models its uncertainty set as $\Omega(\beta) = \{\omega \in \mathbb{R}_+^2 : \mathbf{F}\omega \leq \mathbf{h} + \mathbf{G}\beta\}$, with

$$\mathbf{F} = \begin{bmatrix} \mathbf{f}_1^\top \\ \mathbf{f}_2^\top \end{bmatrix}, \quad \mathbf{h} = (10,\ 10)^\top, \quad \mathbf{f}_i \in \mathbb{R}^{10}, \text{ and } \begin{bmatrix} \mathbf{g}_1^\top \\ \mathbf{g}_2^\top \end{bmatrix}, \quad \mathbf{g}_i \in \mathbb{R}^{10}, \text{ where}$$

$$\mathbf{f}_1 = (0.3372,\ 0.1766,\ 0.4441,\ 0.1798,\ 0.7629,\ 0.3143,\ 0.5914,\ 0.7974,\ 0.7084,\ 0.3197),$$
$$\mathbf{f}_2 = (0.6875,\ 0.3057,\ 0.6642,\ 0.3447,\ 0.6702,\ 0.8915,\ 0.7391,\ 0.2277,\ 0.4021,\ 0.1841).$$
$$\mathbf{g}_1 = (1.7676,\ 0.5272,\ 0.8323,\ 1.4865,\ 0.5928,\ 1.0246,\ 0.9543,\ 1.4439,\ 0.9461,\ 1.7757),$$
$$\mathbf{g}_2 = (1.3427,\ 1.3558,\ 0.5938,\ 1.5645,\ 0.5478,\ 0.7650,\ 1.6331,\ 0.4311,\ 1.0071,\ 0.7519).$$

Performance is measured over 10 independent instances, each with 100 new test points sampled from the 10,000-point dataset. Second-stage results are further averaged over 10 realizations of $\omega$.

As shown in Table 6, our analysis still holds. The DD classifier yields significantly lower second-stage loss (4.37 vs. 25.49) and manipulations (0.29 vs. 17.09) than the DI baseline. This gain arises from anticipating the effect of decisions on manipulation costs, thereby making manipulation more difficult ($\omega_i < 1,\ \forall i$). As shown in Figure 4, $\Omega(\beta^{\mathrm{DD}})$ is strictly smaller than $\Omega(\beta^{\mathrm{DI}})$, which increases second-stage manipulation costs.

As shown in Table 6, our analysis still holds. The DD classifier yields significantly lower second-stage loss (4.37 vs. 25.49) and manipulations (0.29 vs. 17.09) than the DI baseline. This gain arises from anticipating the effect of decisions on manipulation costs, thereby making manipulation more difficult ($\omega_i < 1,\ \forall i$). As shown in Figure 4, $\Omega(\beta^{\mathrm{DD}})$ is strictly smaller than $\Omega(\beta^{\mathrm{DI}})$, which increases second-stage manipulation costs.

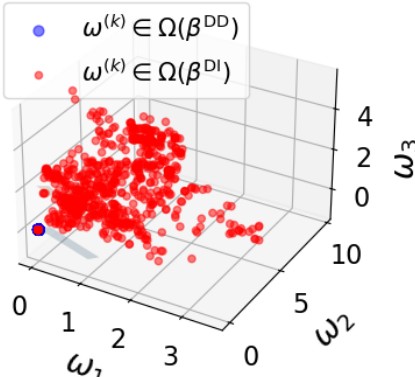

*Figure 4.* Comparison of 3-dimensional projections of the 10-dimensional uncertainty sets $\Omega(\beta^{\mathrm{DD}})$ and $\Omega(\beta^{\mathrm{DI}})$.

*Table 6.* Average performance (mean $\pm$ standard error) across stages and totals, comparing decision-dependent-aware and decision-dependent-unaware classifiers under a diagonal cost matrix for a 10-dimensional setting.

| Metric | First-stage DD | First-stage DI | Second-stage DD | Second-stage DI | Total DD | Total DI |
|---|---|---|---|---|---|---|
| 0-1 Loss | $38.30 \pm 1.70$ | $26.40 \pm 1.43$ | $4.37 \pm 0.04$ | $25.49 \pm 1.62$ | **42.67** | 51.89 |
| Manip. | $33.50 \pm 1.54$ | $26.10 \pm 1.31$ | $0.29 \pm 0.06$ | $17.09 \pm 1.54$ | **33.79** | 43.19 |
| Qualified Manip. | $12.40 \pm 1.17$ | $2.60 \pm 0.60$ | $0.20 \pm 0.04$ | $6.32 \pm 0.62$ | **12.24** | 8.92 |
| Unqual. Manip. | $20.40 \pm 1.17$ | $22.80 \pm 1.38$ | $0.05 \pm 0.02$ | $10.39 \pm 1.01$ | **20.45** | 33.19 |

### E.4. Sub-optimality gap analysis

We have conducted four settings that isolate the impact of key approximations and relaxations in both the first- and second-stage problems. We also vary the McCormick envelope upper bounds to assess their effect on the optimality gap. All experiments were obtained under the same set-up as the main experiment in Section 6, specifically, the same first-stage cost specifications and uncertainty sets.

Table 2 in the main text summarizes the bounds, gaps, and runtimes; here we provide additional discussion and interpretation of these results. The fully linearized formulation yields an average gap of 24.5%. Using a nonlinear first-stage with a dual-norm approximation in the second stage reduces the gap to 19.75%, while removing the dual-norm approximation in the first stage further reduces the gap to 10.75%. This indicates how reducing the approximation and linearization in the first stage reduces the optimality gap and achieves a better solution. It is also worth noting that for this formulation of the nonlinear first-stage, the computational time slightly increases compared to the fully linearized model (11.76 s vs. 13.95 s and 14.63 s). This is because this has been solved using the same solution approach as the linearized model ("Benders C&CG). Finally, when both stages are nonlinear and only the second stage uses the dual-norm approximation, the gap nearly vanishes, albeit at a significantly higher computational cost (approximately 3 hours using brute-force search).

Moreover, the choice of McCormick envelope upper bounds has a significant but non-monotonic impact on the optimality gap. For instance, under upper bounds of $(1, 0.5)$, the gaps are 8%, 5%, and 2% for the fully linearized, dual-norm approximated, and exact nonlinear models, respectively. However, tighter bounds do not necessarily lead to improved performance; for example, under $(0.2, 0.2)$, the gap increases to 25%, 23%, and 20%, respectively. Similarly, under looser bounds of $(2, 1)$, the gaps increase to 44%, 35%, and 10%. These results highlight that the effectiveness of the bounds depends critically on their calibration, and careful tuning is required to achieve high-quality solutions.

In particular, these upper bounds in the McCormick envelope are for $t_2^q$ and $t_2^\omega$, denoted by $t_{2,\mathrm{max}}^q$ and $t_{2,\mathrm{max}}^\omega$, respectively. Intuitively, $\|\Sigma(\omega)^{-\frac{1}{2}}\|_\infty$ characterizes the most cost-efficient direction of manipulation, while $\|\beta'\|_\infty$ measures the sensitivity of the classifier; their product therefore captures the worst-case strategic impact on the firm's loss. Therefore, these bounds

are critical and are typically derived from domain knowledge for obtaining a tight linear approximation.

On this hardware, training the decision-dependent classifier on the 10,000-point dataset required approximately an average of 13.5 seconds per run. By contrast, solving the brute-force benchmark to validate our approximation required about 17,568 seconds (4.88 hours). Therefore, it is computationally feasible to provide this analysis for 13 $((3 \times 4) + 1)$ representative experiments rather than for hundreds or thousands.

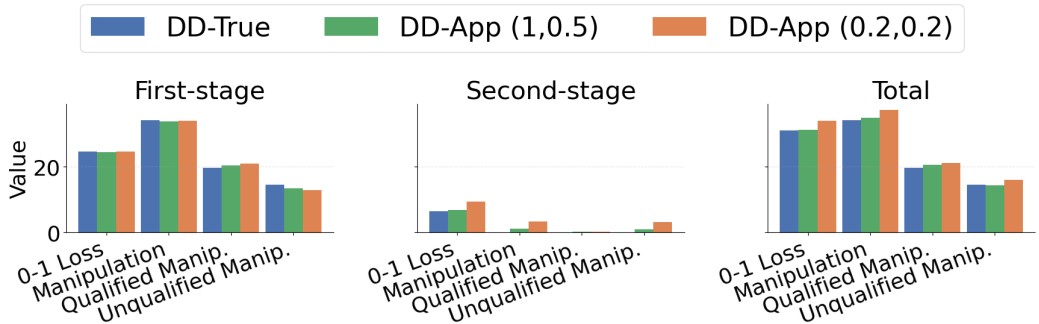

*Figure 5.* Performance Comparison: True dependence aware (DD-True) and our dependence aware (DD-App).

### E.5. Impacts of approximations on reducing uncertainty and curbing manipulations

The optimality gap analysis focused on differences in the overall objective function. Here, we further highlight how our approximate classifier fares against the optimal solution in terms of reducing uncertainty and curbing manipulations.

To this end, we contrast the approximate dependence-aware classifier (DD-App), obtained via our proposed method and under two different McCormick envelope upper bounds, specifically, (1, 0.5) and (0.2, 0.2), against the true optimal dependence-aware classifier (DD-True), computed via brute-force search for the original problem in (10). All experiments use the same experimental set-up as the main experiment in Section 6.

As shown in Figure 5, all classifiers exhibit nearly identical performance in the first stage. However, DD-True achieves slightly better overall performance, primarily due to its superior second-stage outcomes, namely, lower loss and zero manipulation. This suggests that DD-True is more robust to future strategic behavior and better accounts for downstream effects. It is worth noting that, under an appropriate choice of McCormick envelope upper bounds, specifically (1, 0.5), the classifier achieves performance nearly identical to DD-True, further highlighting the importance of careful parameter selection.

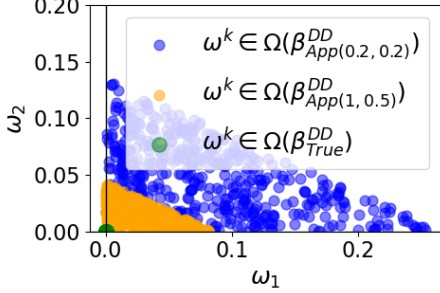

*Figure 6.* Comparison of uncertainty sets $\Omega(\beta_{\text{True}}^{\text{DD}})$ and $\Omega(\beta_{\text{App}}^{\text{DD}})$.

This observation is consistent with the stricter uncertainty set for the DD-True classifier illustrated in Figure 6. Importantly, the overall performance gap remains small, indicating that our approximation is highly effective in learning a classifier that captures the impact of its decisions on future manipulation costs and strategic responses.

## E.6. Additional numerical experiment on synthetic datasets

We generate a synthetic dataset with two-dimensional features $x \in \mathbb{R}^2$, sampled uniformly from $[-10, 10]$. Labels are assigned according to the linear boundary $x_1 + 5x_2 = 2$; specifically, $y = 1$ if $x_1 + 5x_2 > 2$, and $y = -1$ otherwise. To model noisy environments, labels are flipped according to *feature-dependent noise* with probability $\mathbb{P}_{\text{noise}}(x) = 0.5 \exp\left(-\left(\frac{x_1 + 5x_2}{5}\right)^2\right)$, so that points closer to the boundary are more likely to be mislabeled. The dataset consists of 5000 points.

Agents receiving negative classification outcomes may strategically manipulate their features to achieve a positive outcome, gaining benefit $u = 1$ in both stages. The manipulation cost is defined via the $\ell_2$-norm with a non-decreasing function $\phi(r) = 0.1r$.

**First-stage cost.** The cost matrix is fixed and given by $\Sigma_0 = \text{diag}(2, 5)$, where manipulation of feature $x_2$ is 2.5 times more expensive than manipulating $x_1$. The corresponding cost function is:

$$c(x, \hat{x}) = \phi\big(\|\Sigma^{1/2}(\hat{x} - x)\|_2\big) = 0.1\|\Sigma_0^{1/2}(\hat{x} - x)\|_2.$$

**Decision-dependent uncertainty.** To construct the decision-dependent uncertainty set, we fix the interaction matrix $\mathbf{F}$ to be constant and independent of the first-stage decision $\beta$. As discussed in Section 3, this choice ensures that the first-stage decision $\beta$ only influences the *bounds* of the uncertainty set via $\mathbf{G}\beta$, rather than altering the relative interaction between $\omega_1$ and $\omega_2$. The resulting decision-dependent uncertainty set is therefore $\Omega(\beta) = \{\omega \in \mathbb{R}^2_+ : \mathbf{F}\omega \leq \mathbf{h} + \mathbf{G}\beta\}$, with

$$\mathbf{F} = \begin{bmatrix} 1 & 2 \\ 3 & 1 \end{bmatrix}, \quad \mathbf{h} = \begin{bmatrix} 1 \\ 1 \end{bmatrix}, \quad \mathbf{G} = \begin{bmatrix} 5 & 2 \\ 1 & 5 \end{bmatrix}.$$

The diagonal entries of $\mathbf{G}$ are chosen larger than the off-diagonal entries, reflecting that $\beta_i$ primarily governs the cost of manipulating its corresponding feature $x_i$, rather than $x_j$ with $i \neq j$. Finally, we set $\mathbf{h} = (1, 1)^\top$ to capture average manipulation cost, which in the university admission analogy corresponds to the average expense of accessing cheating resources (e.g., buying leaked exams).

For instance, $\omega_1 + 2\omega_2 \leq 1 + 5\beta_1 + 2\beta_2$ and $3\omega_1 + \omega_2 \leq 1 + \beta_1 + 5\beta_2$, where the right-hand sides directly scale with $\beta$. Thus, higher positive weights on particular features expand the feasible set of manipulation costs, reflecting that strategic agents can more easily exploit heavily weighted features.

**Second-stage cost.** The second-stage classifier faces the cost $c(x, \hat{x}) = 0.1\|\Sigma(\omega)^{1/2}(\hat{x} - x)\|_2$, where $\Sigma(\omega) = \text{diag}(g(\omega_1), g(\omega_2)) \cdot \Sigma$, $g(\omega) = \frac{1}{\omega^2}$. Smaller $\omega_i$ increase $g(\omega_i)$, raising the manipulation cost of feature $i$. Together with $\Omega(\beta)$, if $\beta_i > 0$ (feature valued positively by the classifier), the greater the $\beta_i$, the $\omega_i$ has a larger feasible upper bound, making it cheaper to manipulate when $\omega_i > 1$. Conversely, if $\beta_i < 0$, the greater the $\beta_i$ in the negative, the $\omega_i$ tends to have a smaller upper bound, increasing the manipulation cost.

In our experiment, we are not restricting the value of $\beta$ and $\beta'$, meaning we don't have any of $\mathbf{A}\beta \geq \mathbf{b}$, and $\mathbf{B_2}\beta' \geq \mathbf{d} - \mathbf{B_2}\beta - \mathbf{E}\omega$.

We implemented the Benders C&CG algorithm using Python and GurobiPy. Results show that the decision-dependent (DD) classifier alters manipulation costs in the second stage in a way that improves accuracy compared to its decision-independent (DI) counterpart.

*Table 7.* Average $\pm$ standard error across stages and totals.

| Metric | First-stage DD | First-stage DI | Second-stage DD | Second-stage DI | Total DD | Total DI |
|---|---|---|---|---|---|---|
| 0-1 Loss | $25.36 \pm 1.01$ | $22.80 \pm 0.91$ | $5.22 \pm 0.21$ | $43.92 \pm 3.92$ | **30.58** | 66.72 |
| Manipulations | $25.24 \pm 1.02$ | $22.80 \pm 0.78$ | $1.42 \pm 0.45$ | $43.66 \pm 3.91$ | **26.66** | 66.46 |
| Qualified Manip. | $2.80 \pm 0.42$ | $3.80 \pm 0.35$ | $0.60 \pm 0.21$ | $8.53 \pm 2.17$ | **3.40** | 12.33 |
| Unqualified Manip. | $22.44 \pm 0.94$ | $18.92 \pm 0.69$ | $0.82 \pm 0.24$ | $35.11 \pm 3.58$ | **23.26** | 54.03 |

As reported in Table 7, the second-stage average total 0–1 loss for the DD classifier is significantly lower than that of the DI baseline. This stems from the DD classifier's design, which explicitly considers how first-stage decisions affect later manipulation costs. Specifically, Concretely, the average second-stage 0–1 loss under DD is $5.22$, compared to $43.92$ for DI. Similarly, the average total manipulations are $1.42$ under DD and $43.66$ under DI. Among manipulated agents, the DD classifier accepts only $0.6$ qualified and $0.82$ unqualified individuals, versus $8.53$ and $35.11$ under DI.

This reduction occurs because the DD classifier anticipates the effect of its decisions, making manipulation costlier, thereby making manipulation more difficult. Figure 7a illustrates this: $\Omega(\beta^{\mathrm{DI}})$ contains much larger values of $\omega$ than $\Omega(\beta^{\mathrm{DD}})$, reflecting cheaper manipulation under DI. Moreover, Figure 7b shows that $\omega \in \Omega(\beta^{\mathrm{DD}})$ is always strictly less than 1, indicating that the second-stage manipulation cost is strictly higher than in the first stage.

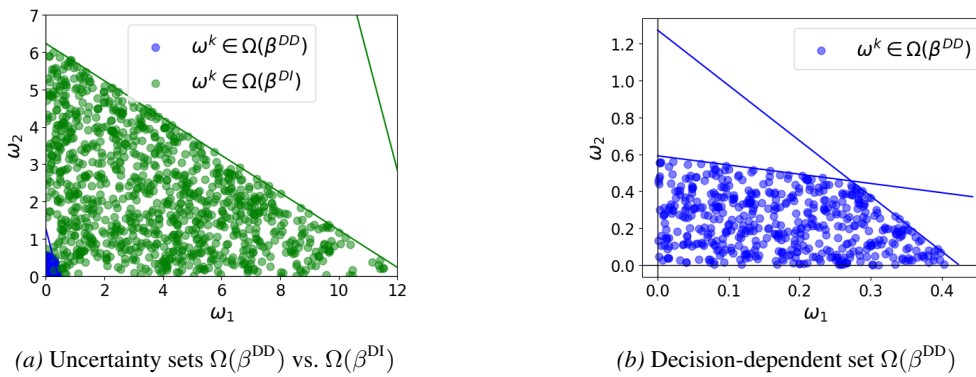

*(a)* Uncertainty sets $\Omega(\beta^{\mathrm{DD}})$ vs. $\Omega(\beta^{\mathrm{DI}})$      *(b)* Decision-dependent set $\Omega(\beta^{\mathrm{DD}})$

*Figure 7.* Second-stage decision-dependent uncertainty sets.

Thus, it both limits the reduction in second-stage costs and ensures manipulation remains consistently expensive.

In contrast, Table 7 also reveals that the DD classifier performs worse in the first stage. Its average 0–1 loss is $25.36$, compared to $22.80$ for DI classifier. This trade-off arises because the DD classifier deliberately sacrifices first-stage accuracy to mitigate second-stage manipulation, whereas the DI classifier—being unaware—optimizes solely for immediate outcomes. For instance, under DD, the average total manipulations reach $25.24$ ($22.44$ unqualified and $2.8$ qualified agents), whereas DI yields $22.80$ manipulations ($18.92$ unqualified and $3.80$ qualified).

Overall, the dependency-aware model achieves lower averages across both stages compared to the unaware model. Specifically, the total 0–1 loss drops to $30.58$ under DD, versus $66.72$ under DI. Similarly, overall manipulations are reduced to $26.66$, compared to $66.46$ for DI. These results confirm that incorporating decision-dependent manipulation costs effectively lowers both classification error and manipulation rates.

