# OpenReview forum: "Robust Strategic Classification under Decision-Dependent Cost Uncertainty"
_ICML.cc/2026/Conference — ICML 2026 regular_

### Official Review · Reviewer_THbN · 2026-03-07

**Soundness:** 4
**Presentation:** 4
**Significance:** 3
**Originality:** 3
**Overall Recommendation:** 5
**Confidence:** 4

**Summary:**

This paper essentially considers a two-stage robust optimization problem, where the first-stage problem is a decision-aware loss that minimizes a joint objective of accuracy and optimization quality for a linear classifier, and the second-stage problem minimizes the same loss with its objective and constraint set altered by an uncertain parameter $\omega$, that is eventually controlled by considering the worst-case as is common in the robust setting. While this is a rather broad setting, the paper does consider a very specific loss function that is used in both time periods, and a very specific manner in which the uncertainties affect the second-stage problem. Because of this more specific structure, the authors adopt a bound for the loss function, which is bilinear in two norms. Based on this, the authors propose optimizing for the bound instead, leveraging on the tractability of linearizing the bilinear terms through McCormick-type tricks. They illustrate their model on the university admissions problem.

**Compliance With Llm Reviewing Policy:**

Affirmed.

**Final Justification:**

I did not update my scores as the rebuttal from the authors has not addressed the core of my concern. Nonetheless, I was initially happy with the work (recommendation = 5) and will keep it as such.

**Key Questions For Authors:**

-

**Limitations:**

Yes – the authors qualify that their work is purely theoretical.

**Strengths And Weaknesses:**

Soundness: I did not look too deeply, but sounds good.

Presentation: Perhaps a bit more discussion of alternatives and why they are avoided would be helpful. See later discussion.

Significance: First off, I would say that decision-driven ambiguity / uncertainty in robust optimization is a known difficult problem, so, as a researcher in robust optimization, I would like to applaud efforts and exploration in this area. Broadly speaking, I have some concerns for multiple aspects of this work, however, considering the general difficulty level of decision-driven uncertainty, I suppose these concerns are something I can live with. First, some type of prediction-based loss for __point predictions__ is chosen, though one might contest the potential stochastic rather than deterministic setting for the decision problem, which definitely will play a part due to the non-linearity in the cost function. The consequence of this is that decisions will surely be binary, whereas a mixed strategy would be technically possible in the stochastic setting. Moreover, there has been significant developments in the contextual optimization area, which is also explored from the robust optimization angle, that might call into question whether the choice of the decision-aware loss is necessarily a bit too simplistic. Adding to this matter, the numerical simulations do not compare against the state-of-the-art in these areas. Second, the specific structure of the decision-dependent uncertainty set $\Omega(\beta)$ could be debated. In the very least, the authors opted for a parameter uncertainty, while there are papers in the literature that have moved on to distributional uncertainty in $\omega$. (9) also feels similar to the uncertainty set structure in other works on decision-dependent uncertainty. Adding these things together, one has to question on what is the level of contribution of the paper to the decision-dependent uncertainty literature.

Originality: While the problem formulation might sound novel in the strategic agents community (I am unfamiliar with that community), its relative contribution to the robust optimization literature is much more limited. The use of dual norms is commonly seen in robust optimization and naturally arises from the structure of duality and even McCormick-type tricks have also been seen specifically in the decision-dependent uncertainty literature.

---

> ### Author Rebuttal · Authors · 2026-03-31
>
> ## Thank you for your feedback
>
> We are grateful for your supportive feedback, and especially for your insights from the robust and contextual optimization perspectives.
>
> &nbsp;
>
> ## On the choice of point predictions vs. stochastic loss, and connections to the contextual optimization area
>
> > First, some type of prediction-based loss for point predictions is chosen, though one might contest the potential stochastic rather than deterministic setting for the decision problem [...]. Moreover, there has been significant developments in the contextual optimization area, which is also explored from the robust optimization angle, that might call into question whether the choice of the decision-aware loss is necessarily a bit too simplistic. Adding to this matter, the numerical simulations do not compare against the state-of-the-art in these areas.
>
> **Response**
>
> While our choice of the point prediction is in line with the standard learning objective considered in the strategic learning literature, we appreciate your insight on the power of moving to the stochastic setting to enable mixed strategies and believe it could be considered in future extensions of our (and others') study of this problem to enable new insights.
>
> We are grateful for your pointer to the contextual optimization area. The most commonly studied frameworks of contextual robust optimization consider the context to be *exogenous* (in contrast to our decision-dependent/endogenous context), though recent surveys of the field (e.g., Sadana et al., European Journal of Operations Research, 2025) highlight "endogenous uncertainty" as a relatively sparse but active research direction. We believe this alternative approach to our problem, while beyond our current scope, would be very promising to pursue, and well situated at the intersection of the ML and OR communities. We are again thankful for this pointer.
>
> &nbsp;
>
> ## On the choice of the decision-dependent uncertainty set, and contribution to the decision-dependent uncertainty literature
>
> > Second, the specific structure of the decision-dependent uncertainty set could be debated. In the very least, the authors opted for a parameter uncertainty, while there are papers in the literature that have moved on to distributional uncertainty in $\omega$. (9) also feels similar to the uncertainty set structure in other works on decision-dependent uncertainty. Adding these things together, one has to question on what is the level of contribution of the paper to the decision-dependent uncertainty literature.
>
> **Response**
>
> We thank you for these comments and will make sure to explicitly highlight the following points in our revision.
>
>
> Our construction of the uncertainty set in (9) indeed aligns with standard polyhedral sets,  commonly used in the literature (e.g. Zeng and Wang 2022), as the polyhedral structure facilitates computational tractability of solving two-stage robust optimization problems with decision-dependent uncertainty. For our context of strategic agents, the choice still allows us to consider settings of interest; for instance, we can capture how additional focus on a feature in stage 1 will (linearly) scale the cost of that feature up or down in stage 2, with the range of this scaling drawn from the polyhedral uncertainty set. We thank you for pointing out the possibility of moving to more general distributional uncertainties, and we will be sure to highlight and explore this as a promising extension of our model.
>
> Our primary contributions are in the learning with strategic agents' community. To the best of our knowledge, while some existing works have considered exogenous and single-stage uncertainty in strategic classification (e.g., Rosenfeld and Rosenfeld (ICML 2024), Ahmadi et al. (ACM EC 2021), Shao et al. (NeurIPS 2023), Tang et al. (AISTATS 2021)), our work is the first in the literature to consider endogenous and temporal (forward-looking) uncertainties, and provides new insights into planning and algorithm robustness in strategic agent settings, by building on the theoretical approaches from the decision-dependent uncertainty literature coupled with our problem-specific relaxations (e.g., the norm bound in Lemma 1).

---

> > ### Author Rebuttal · Reviewer_THbN · 2026-04-01
> >
> > I do not agree with the authors' replies on the above two points, hence I will keep my score as it is. On the first point, the contextual optimization area has made much advances since Sadana et al (note here, 2025 reflects the date of publication and not the date of writing, with a difference that tends to be in the scale of years for the optimization field unlike with ML conferences). On the second point, the reply does not answer my question which is about the contributions to the decision-dependent ambiguity set literature in robust optimization. I do not question the contributions of the paper to the strategic agent literature, as I am wholly unfamiliar with that. That said, I am satisfied with the paper as it is.

---

> > > ### Author Response · Authors · 2026-04-08
> > >
> > > Thank you for your continued support of our paper!  We will certainly make sure to explicitly discuss decision-dependent contextual optimization and distributionally robust variants as important and practically relevant future research directions in strategic classification.

---

### Official Review · Reviewer_Vjmz · 2026-03-09

**Soundness:** 3
**Presentation:** 3
**Significance:** 2
**Originality:** 3
**Overall Recommendation:** 3
**Confidence:** 4

**Summary:**

This paper investigates and concretely formalizes what happens when classifier decisions which aim to curb agents' strategic manipulations endogenously shape manipulation costs over time. The paper focuses on formalizing and establishing the importance of considering time evolving decision-dependent uncertainty. The main contribution of this is the formulation of a Two Stage Robust Optimization (TSRO) problem. They then propose specific approximations and reformulations of the TSRO to make the problem more amenable to existing solution methods for linear TSROs. Through numerical experiments they then move to show that this policies uncovered through solving this optimization framework (i.e., policies that are aware of the policy dependent evolution of costs ) reduce overall uncertainty whilst reducing the amount of manipulation over time.

**Compliance With Llm Reviewing Policy:**

Affirmed.

**Key Questions For Authors:**

see weaknesses (framed as questions as they are the parts I am not clear on)

**Limitations:**

Yes

**Strengths And Weaknesses:**

Strengths:
- Overall, this paper tackles an interesting consideration for strategic classification. Understanding how current decisions impact the optimization problem in the future is an important avenue of research and this viewpoint tackling / thinking about the impact on future costs is interesting.
- The paper is in general well written (though they may be instances here and there where points could be made more clearly) but in general, one is able to understand what the authors are trying to articulate as well as their key contributions. They also do not attempt to overstate they contributions and clearly show where they are building upon previous methods and how their work adds to current understanding of strategic classification.

Weaknesses (and questions):
- My current reading of the scope of their consideration is that they only think about a 2 stage problem (i.e., classifier chooses a classifier, then gets response, classifier chooses once more then there is one more response from the environment / agents). How should the reader be thinking about what happens in the long run beyond just the two stages?
- As a consequence what can we say about the long run performance of both the classifier and the agents?
- My current reading is also that the benefit of this approach is validated in an empircal sense, are there any theoretical guarantees that concretely establish the benefit of the TSRO approach?
- [Minor Framing point] While the numerics of the experiments do illustrate the technical phenomenon of interest, I am not entirely convinced of the framing of adjustment of SAT or extracurricular scores as in general being unfavorable. More students preparing for and doing better in the SAT as well having more extracurriculars could under some views be positive. In fact colleges may seek to induce students to "manipulate" their characteristics as that could be seen as in general beneficial.

Minor editorial points
- Equation 7 has a missing parenthesis

---

> ### Author Rebuttal · Authors · 2026-03-31
>
> ## Thank you for your feedback
>
> We are grateful for your feedback, especially on the problem importance and clarity of our claimed contributions, and questions on long-term effects, theoretical guarantees, and framing (and noting our typo in Eq. (7)!).
>
> &nbsp;
>
> ## Beyond two stages
>
> > How should the reader be thinking about what happens in the long run beyond just the two stages? [...] what can we say about the long run performance of both the classifier and the agents?
>
> **Response**
>
> Our work has indeed focused on the two-stage problem, which is common in the robust optimization literature with DDU (e.g., refs in lines 74-75). As you correctly point out, the problem is inherently multistage. It is known that multistage robust optimization problems, even in the absence of decision dependency, are NP-hard (Delage and Iancu 2018) and thus require specialized methodologies. We view the development of novel models and tractable methods for such general multistage settings as an important and promising direction for future research, one that would naturally build upon our proposed framework.
>
>
> In addition, we believe our two-stage approach can still be usefully employed, when viewed as a *receding horizon* control approach, solving the problem repeatedly over a finite but shifting horizon. Short-horizon planning can be computationally simple and effective in the long-run, as the learned classifier re-shapes costs to limit future manipulation, which can have lasting effects if the problems are self-similar. We hypothesize that this shorter-horizon approach is better than not considering decision-dependent impacts down the line (as also shown in our tests).
>
> &nbsp;
>
> ## Theoretical guarantees
>
> > My current reading is also that the benefit of this approach is validated in an empirical sense, are there any theoretical guarantees that concretely establish the benefit of the TSRO approach?
>
> **Response**
>
> Thank you for the opportunity to discuss theoretical guarantees and benefits of TSROs/our approach.
>
> A seminal work [Zeng and Wang 2022] showed that solving a **linear** TSRO with decision-dependent uncertainties is no more computationally challenging than solving its decision-independent counterpart. Exact computation algorithms exist for this setting, which scale exponentially with the dimension of the uncertainty parameter in the worst-case, though they converge much faster in practice.
>
> As detailed in the paper, the firm problem in Equation (10) has different forms of **nonlinearities** and requires starting with approximations and reformulations to reduce to a linear form. Our linear reformulation has the same worst-case exponential convergence guarantees. We ran new experiments below, and found that we obtain fast convergence in practice.
>
>  |Feature Dimension|2|3|4|10|30|60|
>  |-|-|-|-|-|-|-|
>  |Time (s)|12.95|15.6|16.1|30|94.45|132.38|
>
> Our relaxations do introduce a suboptimality gap. While recent work has established guarantees on the relaxation gap for simpler robust optimization settings (e.g., inexact column-and-constraint generation schemes with controlled inexactness (Tsang et al. 2023)  and approximation guarantees for min-max-min and \$k\$-adaptability approaches (Kurtz 2024)),  existing studies on robust optimization with decision-dependent or reduced uncertainty primarily focus on tractable reformulations and algorithmic convergence (e.g., Zeng and Wang 2022, Chen et al. 2023, Arslan and Poss 2024, Omer et al. 2024). To our knowledge, explicit theoretical guarantees in this setting remain unexplored, requiring substantial new theoretical developments. While interesting, this is beyond our current scope.
>
> &nbsp;
>
>
> ## On Framing of the strategic behavior type
>
> > [Minor Framing point] [...] I am not entirely convinced of the framing of adjustment of SAT or extracurricular scores as in general being unfavorable [...]
>
> **Response**
>
> We thank the reviewer for the opportunity to re-frame and clarify our terminology. Your insight that "colleges may seek to induce some students to manipulate" indeed holds in our findings, where our decision-dependent classifier benefits from inducing qualified students to manipulate. In this context, "manipulation" of SATs could refer to hiring a tutor to learn test tricks (good manipulation) or cheating on them (bad manipulation). In either case, the SAT score changes (feature) but the ability to successfully graduate doesn't (true label). The literature on strategic classification, like us, has taken to refer to this response as "manipulation", as opposed to "improvement", which entails genuine studying and learning to both improve SAT scores (features) *and* the ability to graduate (true label). Incorporating both "manipulation" and "improvement" actions is gaining traction in the strategic ML literature (refs. in lines 58-61), and we find it a natural future extension for our framework, too. We will revisit our writing to reflect these points; thank you.

---

> > ### Author Rebuttal · Reviewer_Vjmz · 2026-04-04
> >
> > The authors have addressed my concerns I will adjust my score.

---

> > > ### Author Response · Authors · 2026-04-08
> > >
> > > Thank you for your support of our paper and for increasing the score!

---

### Official Review · Reviewer_C26L · 2026-03-13

**Soundness:** 3
**Presentation:** 3
**Significance:** 3
**Originality:** 3
**Overall Recommendation:** 5
**Confidence:** 4

**Summary:**

This paper proposes a robust strategic classification method that trains a robust classifier against strategic behaviors, aiming to obtain desirable prediction outcomes. A conventional study in strategic classification typically assumes the cost for feature vectors is fixed. However, in reality, previous prediction outcomes can influence the cost for future manipulations. To address this gap, the authors formulate a learning problem that considers decision-dependent costs to suppress future manipulations. Specifically, they introduce a decision-dependent ambiguity set for the cost matrix used in the Mahalanobis distance, and formulate a two-stage robust optimization problem that minimizes the worst-case loss. Using dual norm relaxation and McCormick envelopes for linearization, the problem is reformulated into a tractable form that can be solved using off-the-shelf optimization solvers and Benders decomposition.

**Compliance With Llm Reviewing Policy:**

Affirmed.

**Final Justification:**

The authors have addressed my concerns. Although I had concerns regarding the unclear scope of applicability for the proposed method, the authors have promised to clarify its limitations in the camera-ready version. Therefore, I have raised my evaluation from Weak Accept to Accept.

**Key Questions For Authors:**

1. I am concerned that the proposed cost matrix model can be too restrictive. Equation (5) introduces the cost matrix in a general form, but aren’t the admissible functions $Q(\cdot)$ and $g(\omega)$ actually quite limited? For example, is $g(\omega)$ restricted only to the form $g_i(\omega_i)=1/\omega_i^2$ (or its positive constant multiples), as used in Appendix E.2? Furthermore, $Q$ must be defined to ensure the resulting cost matrix is positive definite, but it is unclear which specific functions can be used for $Q$. The example of a cost matrix model with off-diagonal components in Appendix E.2 seems quite unnatural, as $\Sigma(\omega)$ relies only on the eigenvectors of $\Sigma_0$ while losing the original eigenvalue information. Could you clarify these limitations?
2. I am interested in the scalability of the proposed method with respect to feature vector dimensionality. Practically, up to how many features can this method handle? Additionally, modern solvers such as Gurobi can handle bilinear terms directly. Is the linearization of bilinear terms via McCormick envelopes necessary? If the problem can be solved directly with bilinear terms, does the McCormick envelope formulation still offer empirical advantages in terms of computational time?

**Limitations:**

yes

**Strengths And Weaknesses:**

Strengths:

- Building prediction models under strategic behavior is an active area of machine learning research. Existing work often assumes overly simplified, fixed user cost functions, ignoring scenarios where the prediction model’s decisions may affect these costs. By formulating a learning problem under a more realistic strategic classification setting, this work provides valuable implications for the machine learning community.
- The mathematical reformulation process is clearly described and technically sound.
- The numerical experiments are comprehensive, and the experimental setups are clearly written in a way that ensures reproducibility.

Weaknesses:

- The conditions under which the proposed reformulation techniques apply are ambiguous. The assumptions regarding the ambiguity set $\Omega(\beta)$ (determined by the model parameter $\beta$) and the covariance matrix model (defined by the parameter $\omega$) are written separately. Consequently, the complete set of assumptions required for the reformulation remains unclear. In particular, the setting for $\Sigma(\omega)$ appears highly restrictive for practical applications. From a robust optimization perspective, a detailed discussion on the design of the cost matrix and the flexibility of its ambiguity set is needed.
- While the proposed reformulation is interesting, the algorithmic contribution is limited. Specifically, the reliance on Benders' decomposition raises significant scalability concerns regarding the dimensionality of the feature vectors.

---

> ### Author Rebuttal · Authors · 2026-03-31
>
> ## Thank you for your feedback
>
> We are grateful for your feedback, especially for your notes on the strengths of our formulation and clarity of theoretical and experimental findings, and your questions on the choice of cost matrices and scalability.
>
> &nbsp;
>
> ## Limitations of cost matrix model
>
> > 1. I am concerned that the proposed cost matrix model can be too restrictive. [...] aren’t the admissible functions \$Q(\cdot)\$ and \$g(\omega)\$ actually quite limited? [...] Furthermore, [...] it is unclear which specific functions can be used for \$Q\$. Could you clarify these limitations?
>
> **Response**
>
> We thank the reviewer for pointing out these important observations. Indeed, the setting for $\Sigma(\omega)$ using the matrix-valued transformation \$Q(\cdot)\$ and mapping function \$g(\omega)\$ must preserve positive definiteness (PD) of the cost-matrix. The choice \$g_i(\omega_i)=1/\omega_i^2\$ in App E.2 ensures this; with other \$g_i(\omega_i)\$, this would require reformulation to be able to write the second-stage problem as an LP first. We will make sure to highlight this limitation.
>
> Similarly, the construction of \$Q(g(\omega))\$ in the off-diagonal examples was one convenient way to ensure that $\Sigma(\omega)$ is PD. More broadly, any mapping \$Q(\cdot)\$ that yields a symmetric positive definite \$\Sigma(\omega)\$ is admissible. We will make sure to clarify these points in our revision, and more explicitly identify choices made possible by our formulation, and their limitations.
>
>
> &nbsp;
>
> ## Scalability; advantage of McCormik envelopes
>
> > I am interested in the scalability of the proposed method with respect to feature vector dimensionality.
>
> **Response**
>
> We thank the reviewer for raising this important point and have run new experiments to illustrate the scalability of our method. Specifically, the number of feature dimensions affects the number of the extreme points in the feasible set of the dual second stage problem. In the *worst case*, the iteration complexity can grow exponentially in the number of extreme points of this feasible set. That said, convergence time in practice is often low. The table below summarizes the time for running our experiments with different feature dimensions.
>
> &nbsp;
>
>  ### Computation Time Across Feature Dimensions
>  |Feature Dimension|2|3|4|10|30|60|
>  |-|-|-|-|-|-|-|
>  |Time (s)|12.95|15.6|16.1|30|94.45|132.38|
>
> &nbsp;
>
> In more detail, you are indeed right that pure Benders decomposition would raise scalability issues, but unlike Benders decomposition, which only adds a constraint, the C&CG component adds both new columns (recourse decisions) and constraints (subproblem scenarios), and by doing so often requires fewer iterations.
>
> While our method remains efficient, evaluating the *ground truth* strategic behavior in higher dimensions is computationally expensive, so we report full experimental results only for the 3D and 4D settings. The tables below show that our findings remain consistent as dimensionality increases.
>
> &nbsp;
>
>  ### 3D Features: Average ± Standard Error
>  |Metric|First-stage DD|First-stage DI|Second-stage DD|Second-stage DI|Total DD|Total DI|
> |-|-|-|-|-|-|-|
> |0-1 Loss|32.72±0.88|25.4±0.66|5.49±0.079|20.14±2.95|**38.21**|45.54|
> |Manipulations|38.76±0.95|32.28±0.64|0.28±0.093|13.00±2.80|**39.04**|45.28|
> |Qualified Manip.|22.84±0.64|8.72±0.46|0.028±0.0085|1.24±0.41|**22.868**|9.96|
> |Unqualified Manip.|15.92±0.76|23.5±0.59|0.248±0.0864|11.70±2.74|**16.16**|35.2|
>
> ### 4D Features: Average ± Standard Error
> |Metric|First-stage DD|First-stage DI|Second-stage DD|Second-stage DI|Total DD|Total DI|
> |-|-|-|-|-|-|-|
> |0-1 Loss|28.2±0.81|16.64±0.78|2.82±0.017|16.88±1.15|**31.02**|33.52|
> |Manipulations|19.88±0.65 | 22.72 ± 0.85 |0.032±0.013|13.79±1.83|**19.91**|36.51|
> |Qualified Manip.|12.24±0.65|8.00±0.47|0.0 ± 0.0|5.78±1.0|**12.24**|13.78|
> |Unqualified Manip.|7.6±0.43|14.72±0.68|0.032±0.013|7.99±0.93|**7.63**|22.71|
>
> &nbsp;
>
> > Additionally, [...] does the McCormick envelope formulation still offer empirical advantages in terms of computational time?
>
> In terms of the use of the McCormick envelope, this is critical for our approach's efficiency. A key step in efficient computation of TSRO solutions given decision-dependent uncertainty is to exploit the fact that the second-stage problem is **linear**, which implies that its dual feasible region is a polyhedron with finitely many extreme points. If bilinear terms are left in their original form, the second-stage problem becomes nonlinear, removing this advantage, and the extreme-point enumeration strategy would no longer apply (see App. D.1.2, Eq. (22)). By restoring linearity, McCormick envelopes allow us to leverage duality and KKT-based reformulations, leading to a tractable sequence of linear programs (see App D.1.3, Eq. (25) and its constraints). We will revise the manuscript to better emphasize this structural role of the McCormick linearization.

---

> > ### Author Rebuttal · Reviewer_C26L · 2026-04-03
> >
> > Thank you for the detailed response. Most of my concerns have been addressed, and I will maintain my positive score.
> >
> > However, the nature of the admissible functions $Q(\cdot)$ remains vague to me.
> > While the authors' claim that any mapping is allowed as long as it preserves positive definiteness is mathematically correct, the specific types of functions admissible as $Q$ in practice remain ambiguous.
> > To demonstrate the practical utility and claimed generality of the model, please provide concrete examples of tractable functional forms for $Q$ (other than the one in Appendix E.2).
> > If providing such examples is difficult, please acknowledge this limitation in the manuscript and note that the framework's applicability may currently be restricted to specific functional structures.

---

> > > ### Author Response · Authors · 2026-04-08
> > >
> > > Thank you for your continued support of our paper! We fully agree with your observations and will make sure to explicitly clarify the limitation of our cost function choice in the revised manuscript.

---

### Official Review · Reviewer_TH84 · 2026-03-14

**Soundness:** 2
**Presentation:** 3
**Significance:** 3
**Originality:** 3
**Overall Recommendation:** 4
**Confidence:** 5

**Summary:**

The paper extends the literature on strategic classification to account for manipulation costs that depend on past algorithmic decisions. To incorporate evolving costs, the paper proposes a two-stage robust optimization (TSRO) framework that uses a decision-dependent uncertainty set. To analyze this framework, the authors present various approximations and reformulations to linearize and relax the problem, following which they use existing solutions for linear TSRO problems. The paper includes numerical experiments to demonstrate the performance of the proposed approach

**Compliance With Llm Reviewing Policy:**

Affirmed.

**Final Justification:**

Authors' responses clarified some aspects such as empirical evalution of the optimality gap and experiments on higher dimensional features. Thus, I raise my score to a weak accept. As I state in my final comments, I would strongly encourage the authors to include these experiments in more detail in the paper. I would also like to see an extended version of the experiments with higher number of features along with the detailed analysis. With these additions, the merits of the paper would outweigh the weaknesses.

**Key Questions For Authors:**

1. What is the theoretical gap between the proposed relaxations and the optimal solution? Alternatively, Is there a way to understand when the proposed relaxations are beneficial vs when they may make this gap large?
2. Could the authors provide further discussion or justification on the choice of the decision-dependent uncertainty set (eq. (9))?

**Limitations:**

yes

**Strengths And Weaknesses:**

**Soundness:** While the paper is well-structured and methodologically introduces the various practical relaxations and the solution, theoretical justification for these choices is largely missing. Specifically, what is the theoretical gap between the proposed method and the optimal solution – this is important in my view to understand and analyze the cases where the relaxation may fail (I acknowledge the authors mention this in the discussion section, however I believe this analysis is important in the current version). Further, the paper claims in Section 5.1 some implications of the asymmetry of the upper bound of the loss: this notion bears similarity to the social cost of strategic classification in [1] and a more formal treatment would be appreciated here given the paper mentions this result as one of the contributions.

I appreciate the numerical experiments with real covariates from university admission datasets. However, the experimental setups with 2 features (for synthetic experiments as well) is limiting in my view, especially as the paper lacks theoretical optimality gap analysis. Expanding the setup to capture real settings with a moderate number of covariates would help in demonstrating the performance of the proposed approach in general settings, especially when features can interact in non-trivial ways.

**Presentation:** The paper is overall well-structured and easy to follow. The description of the problem setup is clear, which makes it easy to understand the subsequent parts. The paper also cites relevant related work and positions their work in the context of past work. There are a few places where the paper is not clear and will benefit from further clarity and complete notation:


- Section 4.2: The dimensions of all variables and functions in eq (9) should be clearly specified in the text.
- The metrics in Table 1 (and all other experimental results) should be formally defined before introducing the results. I did not find this in the paper.

**Significance:** The paper introduces an important problem that occurs in learning settings when machine learning models are deployed for decision-making. A principled and efficient solution to this problem could improve the practical utility of models by making them more incentive-aware in the presence of strategic behavior.

**Originality:** The paper introduces a new and important extension of the strategic classification problem. While the paper combines existing techniques from past work (Rosenfeld
& Rosenfeld, 2024; Zeng & Wang (2022)), this combination is in the service of an original problem. Given the optimality of this combination and intermediate steps are further justified, the contribution would be a strong one.

[1] Smitha Milli, John Miller, Anca D. Dragan, and Moritz Hardt. 2019. The Social Cost of Strategic Classification. In Proceedings of the Conference on Fairness, Accountability, and Transparency (FAT* '19).

---

> ### Author Rebuttal · Authors · 2026-03-31
>
> ## Thank you for your feedback
>
> We are grateful for your feedback, esp. for your comments on the significance and potential for strong originality, and questions on the optimality gap, uncertainty set, and scalability.
>
> &nbsp;
>
>
> ## 1. Optimality gap
>
> > What is the theoretical gap between the proposed relaxations and the optimal solution? Alternatively, Is there a way to understand when the proposed relaxations are beneficial vs when they may make this gap large?
>
> **In short,** while a theoretical analysis of the gap is an open problem and beyond our scope, motivated by your question about alternatives, we conducted new numerical experiments to isolate the impact of each approximation & relaxation.
>
> **Details**
>
> Although we agree that the theoretical analysis of the proposed reformulation is important, such analysis goes beyond our current scope. While recent work has established such guarantees for simpler robust optimization settings (e.g., inexact column-and-constraint generation schemes with controlled inexactness (Tsang et al. '23)  and approximation guarantees for min-max-min and \$k\$-adaptability approaches (Kurtz, '24)), existing studies on robust optimization with decision-dependent or reduced uncertainty primarily focus on tractable reformulations and algorithmic convergence (e.g., Zeng and Wang '22, Chen et al. '23, Arslan and Poss '24, Omer et al. '24). To our knowledge, relaxation gap guarantees in this setting remain unexplored, requiring substantial new theoretical developments.
>
> Given this, we conducted new numerical experiments that isolate the impact of key approximations and relaxations in both the first- and second-stage problems, while also varying the McCormick envelope parameter upper bounds \$t^q_{2,\max}\$ and \$t^\omega_{2,\max}\$ to assess their impact on the gap. All experiments use the same setup as in the main paper.
>
> We consider 4 settings [each followed by resp. observations]: (0) total linearization [avg 24.5% gap]; (1) a linear 2nd-stage with a nonlinear 1st-stage using the dual-norm approximation [avg reduction of 4.75% in gap]; (2) a nonlinear 1st-stage without the dual-norm approximation [avg reduction of 13.75% in gap]; and (3) nonlinear 1st- and 2nd-stages, where only the 2nd-stage uses the dual-norm approximation [gap nearly vanishes but at a much higher cost of ~3 hrs]. Also, proper use of McCormick envelope upper bounds reduces the gap; e.g. avg gap of 5% under (1,0.5) vs. 29.67% under (2,1).
>
> ### Suboptimality Gap (%)
>
> *Settings (0)–(3) are defined above*
>
> | Setting | (\$t^q_{2,\max}, t^\omega_{2,\max}\$)=(0.2, 0.2) | (\$t^q_{2,\max}, t^\omega_{2,\max}\$)=(1, 0.5) | (\$t^q_{2,\max}, t^\omega_{2,\max}\$)=(2, 1) | (\$t^q_{2,\max}, t^\omega_{2,\max}\$)=(0.5, 0.5) | (Avg) Gap | Avg Time |
> |--------|-----------|----------|--------|------------|------|------|
> | (0) | 25% | 8% | 44% | 21% | 24.5% | 11.76 s |
> | (1) | 23% | 5% | 35% | 16% | 19.75% | 13.95 s |
> | (2) | 20% | 2% | 10% | 11% | 10.75% | 14.63 s |
> | (3) No Bounds  |  | - |  | | 0.2216% | 3.11 h |
> | Exact (FS+SS) |  |  - |  |  | 0% | 4.88 h |
>
>
> &nbsp;
>
> ## 2. The uncertainty set
>
> > Could the authors provide further discussion or justification on the choice of the decision-dependent uncertainty set?
>
> **Response**
>
> Thank you for this question. We will clarify this in the paper. We adopt a polyhedral uncertainty set as it is needed to obtain computational tractability in TSRO with DDU. Further, its combination with the second-stage decision-dependent cost matrix in (5) allows us to capture different scenarios of interest for our context. For instance, with the choice of \$g(.)\$ and the matrices in (9), we can capture how additional focus on a feature in stage 1 will (linearly) scale the cost of that feature up or down in stage 2, with the range of the scaling specified by the polyhedral uncertainty set (App. E.1.2. details such g(⋅) & DDU set used in our experiments).
>
> &nbsp;
>
> ## On expanding the experiments
>
> > Expanding the setup to capture real settings with a moderate number of covariates would help in demonstrating the performance of the proposed approach in general settings [..]
>
> **Response**
>
> Thank you for this suggestion. We have now extended our experiments to 3-, 4-, 10-, 30-, and 60-dimensional feature spaces. The first table below shows scalability in increasing dimension.
>
> |Feature Dimension|2|3|4|10|30|60|
> |-|-|-|-|-|-|-|
> |Time (s)|12.95|15.6|16.1|30|94.45|132.38|
>
> Below, we also report full results for the 4D case, which shows that our findings remain consistent with Table 1 in the main paper.
>
> |Metric|First-stage DD|First-stage DI|Second-stage DD|Second-stage DI|Total DD|Total DI|
> |-|-|-|-|-|-|-|
> |0-1 Loss|28.2±0.81|16.64±0.78|2.82±0.017|16.88±1.15|**31.02**|33.52|
> |Manipulations|19.88±0.65 | 22.72 ± 0.85 |0.032±0.013|13.79±1.83|**19.91**|36.51|
> |Qualified Manip.|12.24±0.65|8.00±0.47|0.0 ± 0.0|5.78±1.0|**12.24**|13.78|
> |Unqualified Manip.|7.6±0.43|14.72±0.68|0.032±0.013|7.99±0.93|**7.63**|22.71|

---

> > ### Author Rebuttal · Reviewer_TH84 · 2026-04-03
> >
> > Thank you for your responses.
> >
> > I understand the challenges in computing the optimality gap and appreciate the numerical experiments. I would strongly encourage the authors to include these experiments in more detail in the paper. They play a significant role in establishing the performance of the proposal more formally.
> >
> > I would also like to see an extended version of the experiments with higher number of features along with the detailed analysis.
> >
> > I will raise my score to 4 following the rebuttal (this will be updated with the final justification).

---

> > > ### Author Response · Authors · 2026-04-08
> > >
> > > Thank you for your support of our paper and for increasing the score!
> > >
> > >
> > >
> > > We will certainly make sure to include details of these experiments, including the optimality gap, justification of the decision-dependent uncertainty set, and full results on higher feature dimensions, in the revised manuscript.
> > >
> > >
> > >
> > > For reference, we have extended our experiments to a 10-dimensional setting, as detailed below.
> > >
> > >
> > >
> > >
> > >
> > > ### Results with Higher Dimensions
> > >
> > >
> > >
> > > Performance is evaluated over 10 independent instances, each with 100 test samples drawn from a dataset of size 10,000. Second-stage results are further averaged over 10 realizations of \$\omega\$.
> > >
> > >
> > >
> > > As shown below, our findings remain consistent in higher dimensions. The Decision-Dependent (DD) classifier achieves substantially lower second-stage loss ($4.37$ vs. $25.49$) and manipulations ($0.29$ vs. $17.09$) compared to the  Decision-Independent (DI) baseline. These improvements stem from anticipating how decisions affect manipulation costs, thereby discouraging strategic behavior.
> > >
> > >
> > >
> > > |Metric|First-stage DD|First-stage DI|Second-stage DD|Second-stage DI|Total DD|Total DI|
> > > |-|-|-|-|-|-|-|
> > > |0-1 Loss|38.30±1.70|26.40±1.43|4.37±0.04|25.49±1.62|**42.67**|51.89|
> > > |Manipulations|33.50±1.54|26.10±1.31|0.29±0.06|17.09±1.54|**33.79**|43.19|
> > > |Qualified Manip.|12.40±1.17|2.60±0.60|0.20±0.04|6.32±0.62|**12.24**|8.92|
> > > |Unqualified Manip.|20.40±1.17|22.80±1.38|0.05±0.02|10.39±1.01|**20.45**|33.19|

---

### Decision · Program_Chairs · 2026-04-30

**Decision:**

Accept (regular)

**Comment:**

This work presents and studies an interesting variation on strategic classification in which the learner faces two rounds of user responses: a first (standard) round with known costs, and a second round where costs change based on first round classifier within some uncertainty set. The goal is to jointly learn a pair of robust classifiers for both rounds. The authors propose using a two-stage robust optimization approach and analyze its behavior through a series of transformations and approximations.

Overall the paper does a good job of bringing in ideas from decision-dependent robust optimization to the setting of strategic classification and presents a clean extension of the framework that enables embedding such ideas within it. Some reviewers were unsure about some modeling choices. For some of these, the rebuttal was helpful, and so the authors should include a discussion in the final paper. However, one reviewer suggested several possible alternative directions; the authors are encouraged to better support their choices in light of this feedback.

Several reviewers also raised concerns regarding a lack of theoretical guarantees, and in particular an unclear optimality gap. In the rebuttal the authors partially addressed this via additional experiments; these should be added to the paper, as well as an extended discussion on the difficulty of providing theoretical guarantees, and the potential limitations of the proposed solution in this respect.

A minor comment is that in the abstract the authors present their goal as to "develop robust machine learning algorithms that account for, and reduce, this strategic behavior" - this is somewhat misleading since strategic behavior can also work in the learner's favor, in particular when a positive point is classified as negatives, but by moving to become positive corrects the predictive error.